# Sm-like protein Rof inhibits transcription termination factor ρ by binding site obstruction and conformational insulation

Nelly Said [1], Mark Finazzo [2], Tarek Hilal [1,3], Bing Wang [2], Tim Luca Selinger [1], Daniela Gjorgjevikj[1,5], Irina Artsimovitch [2] ✉ & Markus C. Wahl [1,4] ✉

Transcription termination factor ρ is a hexameric, RNA-dependent NTPase that can adopt active closed-ring and inactive open-ring conformations. The Sm-like protein Rof, a homolog of the RNA chaperone Hfq, inhibits ρ-dependent termination in vivo but recapitulation of this activity in vitro has proven difficult and the precise mode of Rof action is presently unknown. Here, our cryo-EM structures of ρ-Rof and ρ-RNA complexes show that Rof undergoes pronounced conformational changes to bind ρ at the protomer interfaces, undercutting ρ conformational dynamics associated with ring closure and occluding extended primary RNA-binding sites that are also part of interfaces between ρ and RNA polymerase. Consistently, Rof impedes ρ ring closure, ρ-RNA interactions and ρ association with transcription elongation complexes. Structure-guided mutagenesis coupled with functional assays confirms that the observed ρ-Rof interface is required for Rof-mediated inhibition of cell growth and ρ-termination in vitro. Bioinformatic analyses reveal that Rof is restricted to Pseudomonadota and that the ρ-Rof interface is conserved. Genomic contexts of rof differ between Enterobacteriaceae and Vibrionaceae, suggesting distinct modes of Rof regulation. We hypothesize that Rof and other cellular anti-terminators silence ρ under diverse, but yet to be identified, stress conditions when unrestrained transcription termination by ρ may be detrimental.

Bacterial transcription termination factor, ρ, is a hexameric, NTP-dependent 5′–3′ RNA translocase and helicase. A ρ monomer contains N-terminal (NTD) and C-terminal (CTD) domains connected by a flexible linker[1]. The NTD comprises an N-terminal three-helix bundle and a five-stranded β-barrel, the OB-fold, that forms a primary RNA-binding site (PBS); the PBSes are located around one outer rim of the ρ hexamer. The CTDs jointly form a secondary RNA-binding site (SBS) at the center of the ρ hexamer. The ligand-free ρ hexamer adopts an inactive, open-ring conformation, in which it can bind single-stranded RNA (or DNA) at the PBSes[2]. When NTP (preferably ATP) binds to pockets located between the ρ subunits and RNA binds to the SBS, the ring closes, trapping RNA in the central pore of the hexamer and

[1]Laboratory of Structural Biochemistry, Institute of Chemistry and Biochemistry, Freie Universität Berlin, Takustr. 6, D-14195 Berlin, Germany. [2]Department of Microbiology and Center for RNA Biology, The Ohio State University, Columbus, OH, USA. [3]Research Center of Electron Microscopy and Core Facility BioSupraMol, Institute of Chemistry and Biochemistry, Freie Universität Berlin, Fabeckstr. 36a, D-14195 Berlin, Germany. [4]Macromolecular Crystallography, Helmholtz-Zentrum Berlin für Materialien und Energie, Albert-Einstein-Str. 15, D-12489 Berlin, Germany. [5]Present address: Department of Medicine, MRC Laboratory of Molecular Biology, University of Cambridge, Cambridge CB2 0QH, United Kingdom. ✉e-mail: artsimovitch.1@osu.edu; markus.wahl@fu-berlin.de

activating the ρ ATPase[3–5]. Binding of pyrimidine-rich nucleic acids to the PBSes promotes the formation of a closed-ring active state[6].

ρ triggers premature release of antisense, horizontally-acquired and untranslated RNAs from transcription complexes, ensuring that useless or potentially harmful RNAs are not transcribed[1]. Productive RNA synthesis requires protection of transcription complexes from ρ. When RNA and protein synthesis are coupled, such as in fast-growing *Escherichia coli*, a translating lead ribosome is the main line of defense against ρ, as it denies ρ access to the transcribing RNA polymerase (RNAP) or nascent RNA[7]. Bacterial RNAs that are translated poorly (e.g., transcripts of xenogeneic cell wall biosynthesis operons) or not at all (e.g., ribosomal [r] RNAs) require dedicated anti-termination mechanisms to avoid being terminated by ρ[1]. Bacteriophages, pervasive mobile elements that are silenced by ρ[8,9], use analogous mechanisms to ensure that their genes are expressed. Anti-termination is mediated by large ribonucleoprotein complexes, e.g., during rRNA synthesis or phage λ gene expression[10,11], single proteins such as RfaH[12,13] or RNAs[14–16], which are recruited to specific sequences or encoded within their target operons and inhibit termination within just one or a few operons.

It is currently unknown how ρ activity is regulated during periods of slow growth or stress, when translation is inefficient and uncontrolled transcription termination may be detrimental. Cellular levels of ρ are kept almost constant[17] through auto-regulation[18], implying that during stress, termination may be kept in check by regulators that reduce ρ activity. Three cellular anti-termination factors have been reported to inhibit ρ; the RNA chaperone Hfq, the Ser/Thr kinase YihE and the small protein YaeO (a.k.a. Rof in *E. coli*). Hfq can directly bind and inhibit ρ by an unknown, and possibly indirect, mechanism[16,19]. YihE forms a complex with ρ and inhibits ρ-RNA interactions[20]. Rof proteins adopt an Sm-like fold, resembling Hfq (Fig. 1a), and also directly bind ρ[21,22]. Rof and bicyclomycin, an antibiotic that counteracts ρ ring closure[6], have similar effects on the synthesis of a small RNA, *sgrS*[23]. NMR-based docking and RNA binding studies suggested that Rof competes with RNA for binding to ρ[21], but the precise mode of the ρ-Rof interaction and of Rof-mediated ρ inhibition remain unclear.

In this work, we use a combination of structural, biochemical and genetic approaches to delineate in detail the mechanism of Rof-mediated inhibition of ρ. We show that Rof reduces ρ-dependent termination in vitro and inhibits *E. coli* growth, presumably by interfering with the essential functions of ρ. We present cryogenic electron microscopy (cryoEM)/single-particle analysis (SPA)-based structures of ρ-Rof complexes, precisely defining the ρ-Rof interfaces and showing that Rof undergoes substantial conformational changes upon binding to ρ. Substitutions of interface residues interfere with Rof activity in vitro and in vivo, confirming that the observed molecular interactions are required for ρ inhibition. Our structure of ρ bound to a 99-nt long RNA that concomitantly occupies the PBSes and the SBS confirms the existence of an extended PBS[24] that is blocked upon Rof binding. Furthermore, structural comparisons show that Rof undercuts functional communication between ρ NTDs and CTDs via the connecting linker. In addition, we show that Rof prevents ρ binding to a transcription elongation complex (EC). Thus, Rof blocks termination by inhibiting ρ recruitment to either the naked RNA or to ECs. We hypothesize that Rof and other known or yet to be identified anti-terminators mediate stress-specific responses by attenuating ρ activity.

## Results
### Rof is toxic in vivo and prevents ρ-dependent transcription termination in vitro
A pioneering study of *E. coli* (*ec*) Rof had shown that, when over-expressed, Rof co-purifies with ρ and induces readthrough of a canonical ρ terminator in the cell; surprisingly, however, purified Rof did not inhibit termination in vitro[25]. We wondered if the presence of an

N-terminal His-tag could have been responsible for this apparent loss of inhibition. The cellular function of Rof is not yet known, prompting us to establish a proxy assay for Rof activity in vivo. We previously showed that ectopic expression of Psu, an unrelated phage P4 inhibitor of ρ, directed by an IPTG-inducible $P_{trc}$ promoter was toxic[26]; we, thus, constructed a plasmid with the *rof* open reading frame (ORF) under the control of $P_{trc}$. A plasmid carrying wild-type (wt) *rof*, but not an empty vector, strongly inhibited growth of IA227[27], a derivative of MG1655 (Fig. 1b). Rof-mediated toxicity was abolished by the addition of an N-terminal His-tag or substitution of D14, a conserved Rof residue shown to be critical for the ρ-Rof interaction[21], for alanine (Fig. 1b).

To determine at what stage of cellular growth Rof expression is inhibitory, we monitored the growth kinetics of IA227. We found that Rof[wt] extended the lag phase from 2.75 to 12.65 h but did not alter the maximal growth rate or the saturation point (Supplementary Fig. 1a). The extended lag could reflect defects in protein synthesis required to sustain rapid growth[28,29] or loss of viability of the seed culture in the stationary phase. In support of the second explanation, Rof expression in IA227 strongly reduced viable colony count (Fig. 1b). By contrast, Rof expression was not toxic to wt *E. coli* and *Salmonella typhimurium* strains[23,30]. To determine the cause of Rof toxicity in IA227, we sequenced its genome; we found that IA227 differs from the MG1655 reference genome in positions of mobile elements and several nucleotide polymorphisms (Supplementary Table 1). Among them is an early stop codon in *rpoS*, which encodes a "general stress response" σ[S] factor that orchestrates adaptation to changes in cellular environment and adverse conditions[31]. Our results show that the *rpoS* null sensitizes cells to Rof (Supplementary Fig. 1b), suggesting an interplay between Rof activity and (still unknown) stress.

While our in vivo results (Fig. 1b) support the hypothesis that the failure to observe anti-termination in vitro[25] could be due to the presence of the His-tag on Rof, it is also possible that Rof, proposed to compete with RNA for binding to ρ[21], may not be able to outcompete a tightly bound RNA; the failed in vitro trials[25] used λ *tR1*, one of the highest-affinity ρ ligands known. To test this idea, we compared effects of tag-less Rof in vitro on templates carrying λ *tR1* and a leader region of *Salmonella siiE*, which is inhibited by ρ in vivo but lacks C-rich *rut* elements preferentially recognized by ρ[32]. We found that ρ terminated RNA synthesis on both templates (Fig. 1c), supporting a view that the *rut* site is less important in the context of the transcription complex[33,34]. Tag-less Rof efficiently counteracted ρ-dependent termination on both templates, suggesting that an N-terminal His-tag indeed interferes with Rof activity and arguing that the Rof effects on ρ are direct and independent of *rut* sites (Fig. 1c).

### Rof binds to ρ via a conserved interface
Consistent with pull-down assays[25], we found that Rof forms a stable complex with ρ in analytical size-exclusion chromatography (SEC), as a mixture of both proteins eluted at an earlier volume than either protein alone (Fig. 2a). Multi-angle light scattering (MALS) detection revealed that ρ and the ρ-Rof complex had heterogeneous molecular mass distributions (Fig. 2a), consistent with earlier observations that ρ does not exist as a discrete hexamer in solution[35].

We subjected SEC fractions corresponding to ρ-Rof complexes to cryoEM/SPA in the absence and presence of the transition state analog, ADP-BeF$_3$ (Supplementary Figs. 2–5 and Supplementary Table 2). Multi-particle 3D refinement with ~600,000 particle images yielded two cryoEM reconstructions for each sample (Supplementary Figs. 3 and 5), corresponding to two ρ-Rof assemblies. In the first assembly, ρ forms an open hexameric ring ($ρ_A$-$ρ_F$) bound by five Rof molecules (Rof$_a$-Rof$_e$; Fig. 2b), while the second assembly represents a ρ pentamer ($ρ_A$-$ρ_E$) bound by four Rof molecules (Rof$_a$-Rof$_d$). In the presence of ADP-BeF$_3$, we obtained a ~1:1 ($ρ_6$-Rof$_5$:$ρ_5$-Rof$_4$) distribution between the two classes, whereas in the absence of the nucleotide, the equilibrium was shifted towards the lower oligomeric assembly

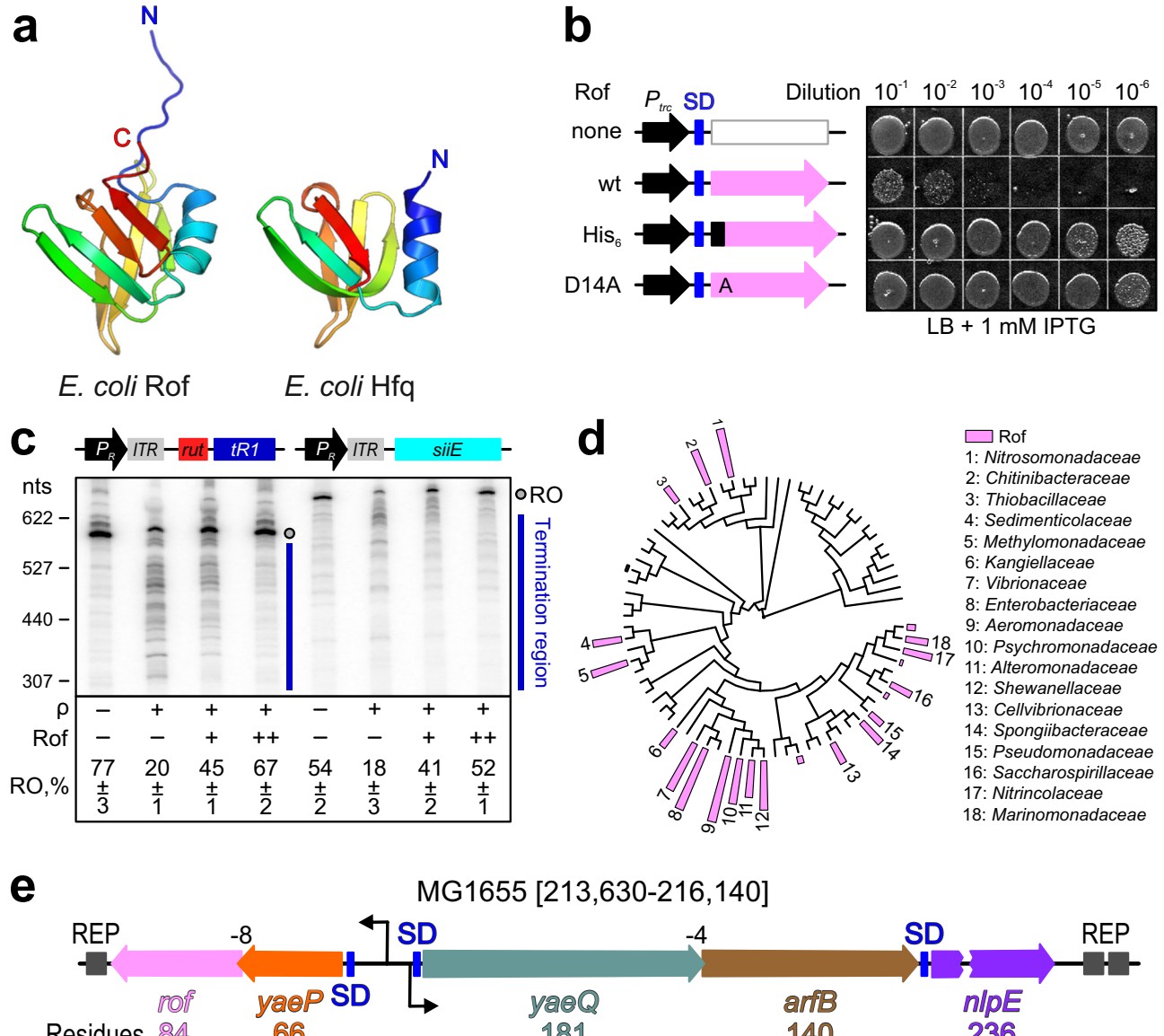

**Fig. 1 | Rof structure, functions, conservation and genetic organization.**
**a** Structural comparison of *E. coli* Rof (left) and Hfq (right). Proteins are shown in rainbow colors from N-termini (N; blue) to C-termini (C; red). **b** Cells transformed with plasmids expressing wild-type (wt) Rof and Rof variants under the control of an IPTG-inducible $P_{trc}$ promoter or an empty vector were grown overnight in LB supplemented with carbenicillin at 32 °C. Serial 10-fold dilutions were plated on LB-carbenicillin in the presence of IPTG and incubated overnight at 32 °C. A set representative of five independent experiments is shown. Shine-Dalgarno (SD) elements are indicated by blue bars. **c** Rof inhibits ρ-dependent termination in vitro. Top, DNA templates encoding the *rut⁺* λ *tR1* (left) or *rut⁺ siiE* leader region (right) downstream of the λ $P_R$ promoter and a C-less cassette that allows for the formation of synchronized, radiolabeled ECs halted 26 nts downstream of the transcription start site. Bottom, representative single-round transcription reactions. Halted ECs

were incubated with ρ and/or Rof (where indicated), restarted by addition of NTPs, and subsequently quenched. Reactions were analyzed on 5% urea-acrylamide gels. Positions of the full-length run-off (RO) RNA and termination regions are indicated, as are the sizes of the molecular weight markers generated by γ³²P-labeling of pBR322 MspI fragments. The fractions of run-off RNA represent the mean ± SD of three independent experiments. Source data are provided as a Source Data file. **d** Rof distribution on the phylogenetic tree of Pseudomonadota. Each leaf represents one family. The height of the bar indicates the fraction of Rof in each family, with the highest one being 80%. **e** *E. coli yae* locus (MG1655 genome coordinates 213,630-216,140); the length of each ORF is indicated below the sequence, the *nlpE* gene is interrupted to save space. The *arfB* and *rof* genes lack SD elements and overlap with the respective upstream ORFs, *yaeP* and *yaeQ*, by 4 and 8 nts. The "insulating" REP elements are shown by dark gray boxes.

($\rho_6$-Rof$_5$:$\rho_5$-Rof$_4$ - 1:3). The local resolutions of the reconstructions decrease towards the regions corresponding to terminal ρ subunits and bound Rof (Supplementary Figs. 2e and 4e). Weaker density for the peripheral Rof molecules could indicate a higher flexibility or a lower occupancy. We built an initial atomic model of ρ-bound Rof based on the density for Rof$_c$ in the $\rho_6$-ADP-Rof$_5$ structure, which is best defined (Fig. 2b, c). This model was used as a starting model for other Rof molecules in the various complexes. The reconstructions do not reveal differences in the ρ-Rof interfaces, irrespective of

oligomeric state or bound nucleotide. We therefore focus our subsequent descriptions on the $\rho_6$-ADP-Rof$_5$ complex and on Rof$_c$ when discussing interaction details.

Rof molecules are bound between protomers $\rho_A$-$\rho_B$, $\rho_B$-$\rho_C$, etc., but not preceding or following the terminal ρ subunits, $\rho_A$ and $\rho_F$ (Fig. 2b). In each case, Rof interacts primarily with one ρ NTD through its N-terminal α-helix (α1, residues 9–23), the loop connecting strands β3 and β4 (residues 46–50) and the very C-terminal region (Fig. 2c). In addition, loop$^{β3-β4}$ of Rof contacts the N-terminal three-helix bundle of

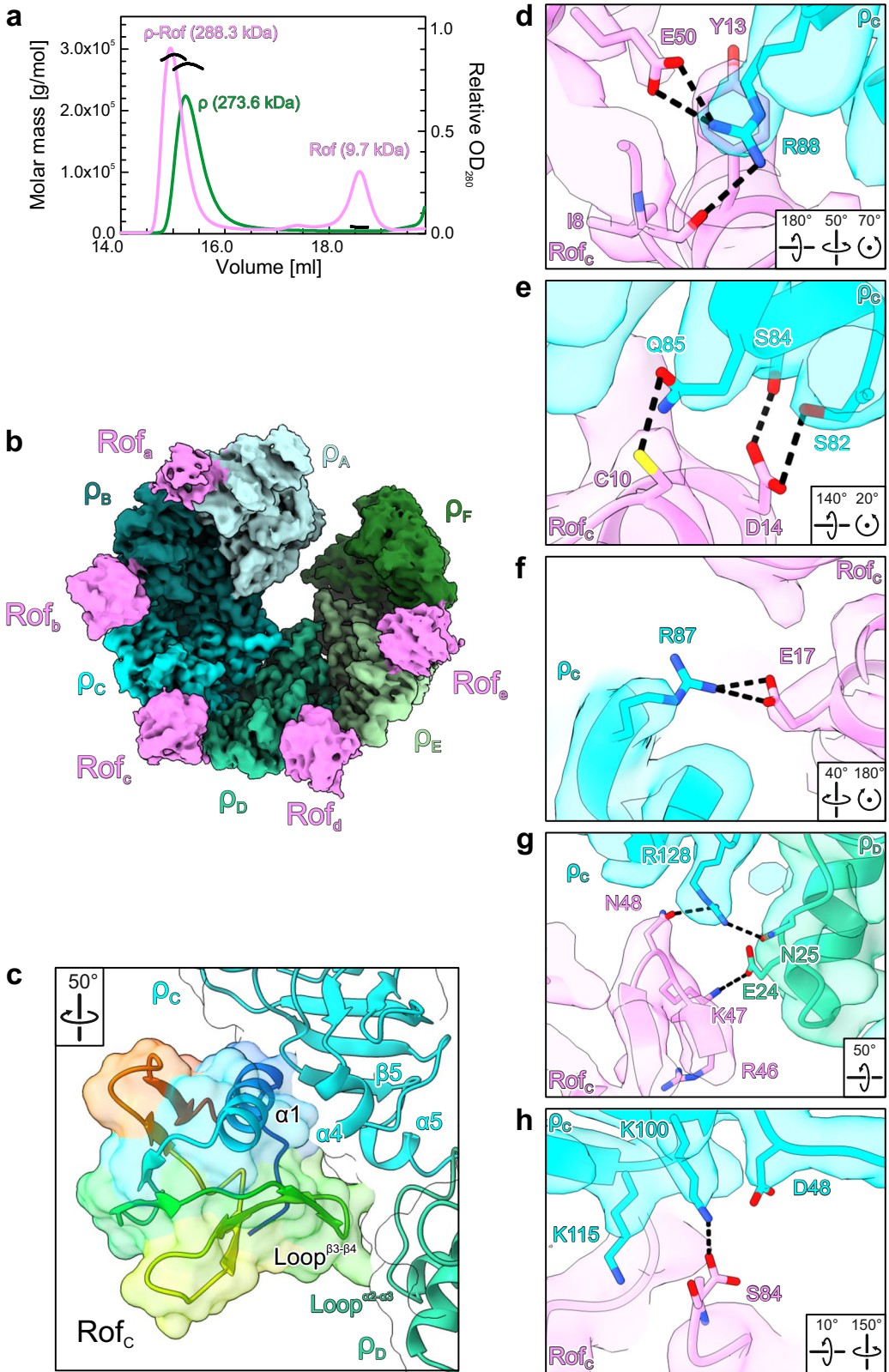

**Fig. 2 | ρ-Rof interaction. a** SEC-MALS of isolated ρ (green) and ρ in the presence of Rof (violet). The average molar masses of the peak fractions are indicated. **b** CryoEM reconstruction of the ρ₆-ADP-Rof₅ complex. ρ, different shades of cyan and green; Rof, violet. ρ adopts an open ring conformation and Rof proteins bind between the NTDs of neighboring ρ protomers. **c** Close-up view of the ρ-Rof interface. Rotation symbols in this and (**d**–**h**) views are relative to (**b**). Rof is shown in rainbow colors from N-terminus (blue) to C-terminus (red). Rof binds ρ via its N-terminal helix α₁, the β3-β4 loop and the C-terminus. **d**–**h** Details of the ρ-Rof interaction. Interacting residues are shown as sticks with atoms colored by type; carbon, as the respective protein subunit; nitrogen, blue; oxygen, red. Black dashed lines, hydrogen bonds or salt bridges. Regions of the ρ₆-ADP-Rof₅ cryoEM reconstruction are shown as semi-transparent surfaces.

the adjacent ρ protomer (Fig. 2c), presumably explaining why Rof binding is only observed between neighboring ρ subunits but not at the terminal subunits of the open ring. The N-terminus and helix α1 of Rof are positioned in a cavity of the ρ NTD between ρ helix α4 (residues 83–89) and one flank of the ρ OB-fold (residues 95–120; Fig. 2c). Close contacts between the Rof N-terminus and ρ could explain why the addition of a His-tag blocks Rof activity in vivo (Fig. 1b). Loop$^{\beta 3\text{-}\beta 4}$ of Rof is positioned atop helix α5 within the ρ NTD-CTD linker (the "connector") and the loop between helix α2 and α3 of the adjacent ρ protomer (residues 22–30; Fig. 2c). The very C-terminus of Rof rests on the exposed surface of the ρ OB-fold next to the Rof N-terminal region (Fig. 2c).

ρ-Rof contacts are predominantly polar (Fig. 2d–h and Supplementary Fig. 6). The backbone carbonyl oxygen of Rof$^{I8}$ (N-terminus) as well as the side chains of Rof$^{Y13}$ (helix α1) and Rof$^{E50}$ (β3-β4 loop) contact ρ$^{R88}$ by hydrogen bond, cation-π and ionic interactions, respectively (Fig. 2d). Rof$^{C10}$ and Rof$^{E17}$ (helix α1) form hydrogen bonds and a salt bridge with ρ$^{Q85}$ and ρ$^{R87}$, respectively (Fig. 2e, f). Rof$^{N48}$ (β3-β4 loop) is hydrogen bonded to ρ$^{R128}$ (connector; Fig. 2g). The C-terminal carboxy group of Rof$^{S84}$ approaches ρ$^{K100/K115}$ (Fig. 2h). Less well-defined interactions include Rof$^{D14}$ (helix α1) hydrogen bonding with ρ$^{S82/S84}$ (Fig. 2e) and Rof$^{K47}$ (β3-β4 loop) interacting electrostatically with ρ$^{E24}$ of the adjacent ρ subunit (Fig. 2g). Most of the ρ-contacting residues of Rof are conserved (Supplementary Fig. 6a). Likewise, Rof-contacting residues of ρ are highly conserved (Supplementary Fig. 6b). As ρ is more widespread among bacteria than Rof, the conservation of the Rof-contacting ρ residues likely reflects their importance in forming the PBS.

## Replacement of ρ-contacting residues undermines Rof function

To validate our structural observations, we replaced Rof residues Y13, D14 or E17 in helix α1 with alanine and tested binding of the corresponding Rof variants to ρ$^{wt}$ in analytical SEC (Fig. 3a). While the ρ interaction was only very mildly affected by an E17A exchange in Rof, ρ binding was completely abolished in Rof$^{Y13A}$ and Rof$^{D14A}$, as monitored by SEC. We also generated ρ variants, ρ$^{R88E}$, ρ$^{F89S}$ and ρ$^{K115E}$, and tested them for Rof$^{wt}$ binding (Fig. 3a). Interaction of ρ$^{R88E}$ with Rof$^{wt}$ was abrogated in SEC, while Rof$^{wt}$ binding was only slightly reduced in ρ$^{F89S}$ or ρ$^{K115E}$ (Fig. 3a). Thus, our structures delineate a precise network of interactions that are crucial for ρ-Rof complex formation.

Rof variants that were unable to bind ρ were also unable to elicit anti-termination in vitro: ρ terminated synthesis of 90% and 86% of transcripts in the presence of Rof$^{Y13A}$ and Rof$^{D14A}$, respectively, as compared to 16% in the presence of Rof$^{wt}$ and 87% in the absence of Rof (Fig. 3b). In agreement with only slightly reduced affinity of Rof$^{E17A}$ for ρ, Rof$^{E17A}$ was only slightly less effective in inhibiting ρ than Rof$^{wt}$ (25% of transcripts terminated in the presence of Rof$^{E17A}$; Fig. 3b). These results indicate that the ρ-Rof contacts observed in our structures are relevant for Rof-dependent inhibition of ρ in the context of complete ECs.

Next, we evaluated the effects of exchanging Y13, D14 and E17 and an additional ρ-contacting residue in helix α1 of Rof, C10, on toxicity in vivo. We measured the lag phase extension upon IPTG induction of plasmid-encoded Rof variants (Fig. 3c). While Rof$^{wt}$ increased the lag 4.6–fold, substitutions of several conserved residues in the N-terminal helix of Rof strongly reduced this effect: plasmids guiding the production of Rof$^{Y13A}$, Rof$^{D14A}$ or N-terminally His-tagged Rof were indistinguishable from the empty vector, whereas Rof$^{C10S}$ (Fig. 2e) increased the lag 2.4-fold. As Rof$^{E17A}$ did not show significant defects in vitro (Fig. 3b), we tested the effects of the Rof$^{E17K}$ variant instead; the charge reversal led to a complete loss of Rof toxicity. By contrast, substitutions in the Rof β3-β4 loop (Fig. 2g) had smaller effects: R46A and K47A exchanges in Rof increased the lag 2.6- and 3.2-fold, respectively, and Rof$^{N48A}$ was almost indistinguishable from Rof$^{wt}$ (Fig. 3c). We conclude that in particular contacts between ρ and the N-terminal helix of Rof

observed in our structures are important for Rof-mediated inhibition of *E. coli* growth.

Substitutions of the surface-exposed Rof residues that contact ρ would not be expected to alter the Rof structure, and thus cellular production or stability. To ascertain that the observed in vivo defects are due to the loss of binding of the Rof variants to ρ, we used Western blotting to quantify Rof variants in cells. As specific anti-Rof antibodies are not yet available, we inserted a single HA tag into a flexible loop of Rof at residue 57, the only location that we found to tolerate tags without the loss of in vivo activity. We found that D14A, E17K, R46A and E47A variants of Rof were produced at levels comparable to, or higher than, Rof$^{wt}$, whereas the Rof$^{Y13A}$ variant was less abundant (Fig. 3d).

## Rof undergoes pronounced conformational changes upon binding to ρ

A solution structure (PDB ID: 1SG5)[21] and an AlphaFold[36] model of isolated *E. coli* Rof showed marked differences and high flexibility in the N-terminal region compared to our ρ-bound Rof. In the isolated state, the N-terminal region lacks contacts to the globular portion of Rof and thus remains flexible, while it is immobilized and caps one face of the OB-fold in the ρ-bound state (Fig. 4). Concomitantly, helix α1 is oriented differently relative to the β-barrel in the isolated and ρ-bound states (Fig. 4). As a consequence, the N-terminal region and helix α1 are repositioned relative to the other main ρ-interacting regions of Rof, the β3-β4 loop and the C-terminus (Fig. 4). The above comparisons show that, upon ρ engagement, Rof undergoes conformational changes and folding/immobilization transitions, which are required to generate the productive ρ-binding surface, composed of N-terminal region, helix α1, β3-β4 loop and C-terminus. In contrast to Rof, no global conformational changes are observed in the ρ protomers upon Rof engagement (e.g. compared to an open-ring, ANPPNP-bound ρ structure; PDB ID: 1PVO)[2].

A model of an *E. coli* ρ-Rof complex has previously been proposed based on the solution structure of Rof and NMR-guided docking[21]. While the general binding site of Rof on the ρ NTDs agrees with our cryoEM structures, the previous docking model differs significantly in detail, as the conformational changes upon binding were not captured. Presently, we cannot exclude that a minor population of isolated Rof adopts a conformation observed in the ρ-bound state, and that binding might, thus, occur via a conformational selection mechanism. However, presently available data favor a mechanism that involves binding via a pronounced induced fit in Rof. Irrespectively, apart from steric hindrance (see above), an N-terminal tag on Rof might also interfere with the conformational changes and folding/immobilization upon binding required for stable complex formation with ρ, as also observed with other proteins[37].

## Rof blocks an extended PBS of ρ

Electrophoretic mobility shift assays indicated that Rof interferes with RNA binding to the ρ PBSes[21]. To further evaluate Rof effects on RNA binding at ρ PBSes, we quantified the affinity of a 5′-FAM-labeled DNA oligomer, dC$_{15}$, that binds exclusively to the ρ PBS[6] by fluorescence anisotropy (Fig. 5a). ρ alone bound dC$_{15}$ tightly ($K_d = 1\,\mu M$; Fig. 5b), in agreement with previous results[38]. ρ binding to dC$_{15}$ was reduced stepwise in the presence of increasing concentrations of Rof$^{wt}$, but not in the presence of Rof$^{Y13A}$ or Rof$^{D14A}$ (Fig. 5b), suggesting that Rof blocks the ρ PBSes or parts thereof.

A structure of ρ bound to a short oligonucleotide at the PBS (PDB ID: 1PVO) had delineated a core PBS that accommodates a dinucleotide[2], and comparison to our ρ-Rof complex structures showed that Rof does not interfere with RNA contacts at the core PBS. However, a recent structure of a pre-termination complex showed that RNA can additionally occupy a region proximal to the core PBS[24], consistent with the observation that longer PBS ligands bind ρ more tightly and have greater impact on ρ ring closure[3,6]. We attempted to

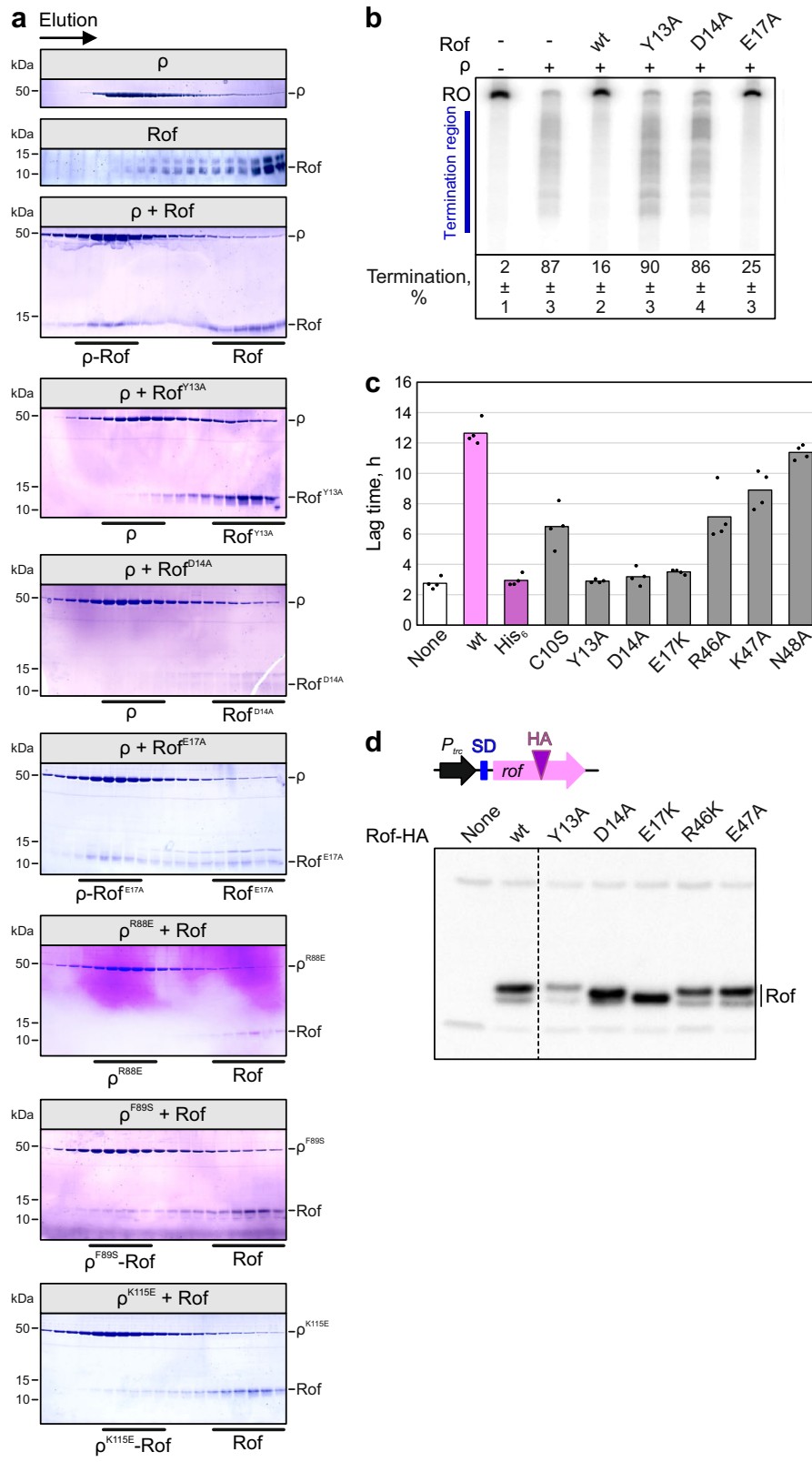

further narrow down the path of RNA along the ρ PBSes by determining a cryoEM/SPA structure of ρ in complex with a 99-nt long, natural *rut* RNA derived from the λ *tR1* terminator, which encompasses both SBS- and PBS-binding regions (Supplementary Fig. 7a). In the presence of ADP-BeF₃, we obtained one reconstruction at a global resolution of 2.9 Å (Supplementary Figs. 8 and 9 and Supplementary Table 2).

In the ρ-*rut* RNA structure, ρ adopts a closed conformation very similar to ρ in complex with an rU₇ SBS ligand and ADP-BeF₃ (PDB ID: 5JJI[3]) and to ρ in a pre-termination complex (PDB ID: 8E6W[24]; Fig. 5c). Density for six nucleotides (nts) is clearly defined at the SBS in the center of the ring (Supplementary Fig. 7b). We tentatively assigned the sequence at the very 3′-end of the RNA ligand to this region (Supplementary Fig. 7a). Density corresponding to RNA regions at the ρ PBSes

**Fig. 3 | Effects of Rof and ρ variants. a** Analytical SEC runs, monitoring the interaction between the indicated ρ and Rof variants. For each run, the same fractions were analyzed by SDS-PAGE. First and second panel, SEC runs of isolated ρ and Rof, respectively. Third panel, binding of ρ^wt to Rof^wt. Panels 4–6, binding of indicated Rof variants to ρ^wt. Panels 7–9, binding of indicated ρ variants to Rof^wt. Experiments were performed three times independently with similar results. Source data are provided as a Source Data file. **b** Effects of Rof variants on ρ-dependent termination in vitro. Assays were done using the λ *tR1* template as in Fig. 1c. Termination efficiency values represent means ± SD of three independent experiments. Source data are provided as a Source Data file. **c** Lag times of growth for IA227 transformed with $P_{trc}$ plasmids expressing different Rof variants or none. $n = 4$ biologically independent samples. Source data are provided as a Source Data file. **d** Production of Rof variants containing a single HA-tag inserted after residue 57 in IA227 cells. Control experiments demonstrated that the expression of HA-tagged Rof is toxic. Rof expression following the induction with 1 mM IPTG was determined by Western blotting with anti-HA antibodies (Millipore Sigma). Experiments were performed three times independently with similar results. Source data are provided as a Source Data file.

is fragmented, suggesting that corresponding RNA regions are dynamically bound and precluding their reliable modeling. Irrespectively, the reconstruction is consistent with six nts bound in an extended conformation (Fig. 5d). No density is observed connecting the PBS-bound RNA regions or between a PBS-bound RNA region and RNA at the SBS (Fig. 5c). At the PBSes, the two 3′-most nts are accommodated at the core PBS as observed before[3]. Density for the preceding nts is lined by ρ^K102, ρ^R105 and ρ^K115 on one side, and by ρ^D60, ρ^F62, ρ^P83, ρ^S84, ρ^Q85, ρ^R87, ρ^R88 and ρ^F89 on the other (Fig. 5e). Consistent with this observation, binding of ρ^R88E and ρ^K115E variants to the λ *tR1 rut* RNA was completely abolished or significantly reduced, respectively, compared to ρ^wt, while binding of ρ^F89S was undisturbed (Supplementary Fig. 7c). These results confirm and further validate an "extended PBS" as observed in a pre-termination complex[24], which augments the RNA affinity and specificity of the ρ core PBS.

Structural comparisons between our ρ-Rof and ρ-*rut* RNA structures showed that bound Rof would block RNA binding at the extended PBS (Fig. 5f). Consistent with the idea that Rof blocks ρ activity by outcompeting RNA at the ρ extended PBS, residues that contribute to the extended PBS are also important for Rof binding (Figs. 2d–h and 5e).

### RNA binding at the extended PBS supports ρ dynamics accompanying ring closure

The classical mechanism of ρ-dependent termination[39], recently visualized by cryoEM[24], involves ATP hydrolysis-dependent 5′–3′ translocation of ρ on RNA. To activate ATP hydrolysis and RNA translocation, ρ has to convert from an open to a closed-ring conformation[3,4]. Ring closure involves conformational changes in ρ that affect the protomer interfaces[3,5]. Upon ring closure, the NTDs of adjacent ρ subunits separate, breaking contacts between the NTD-CTD connector of one ρ protomer and the N-terminal three-helix bundle of the adjacent ρ protomer; simultaneously, the ρ CTDs approach each other more closely, encircling the RNA at the SBS and forming ATPase-active ATP-binding pockets[2,5].

In open ρ structures, ρ^R128 at the C-terminus of connector helix α5 contacts ρ^N25 of a neighboring NTD, helping to position ρ^R28 of the adjacent protomer in a binding pocket that involves helix α5 of the connector (Fig. 6a). These inter-subunit contacts are broken during ring closure; ρ^R128 now hydrogen bonds to the backbone carbonyl of ρ^R88 and/or is engaged by the side chain of ρ^E125, and ρ^R28 is exchanged for ρ^K130 of the same protomer (Fig. 6b). Furthermore, in open and closed ρ structures, the NTD-CTD connector establishes different contacts to the NTD and CTD of its own protomer as well as to the neighboring ρ subunit; upon ring closure, interactions along the entire NTD-CTD connector shift in register by one residue[40,41] (Fig. 6c, top and middle panels). Thus, the NTD-CTD connector establishes dynamic interaction networks, differentially interconnecting the NTD and the CTD within a protomer as well as differentially contacting the neighboring protomer in the open and closed states.

Apart from ATP binding between neighboring CTDs and RNA binding at the SBS, ring closure is also promoted by RNA binding at the PBSes[27,41]. RNA binding at the extended PBS observed in a pre-termination complex[24] and in our ρ-*rut* structure could explain this phenomenon. Residues forming the extended PBS and the 5′-end of the

PBS-bound RNA region are in direct proximity of connector helix α5. We surmise that RNA bound at the extended PBS supports the closed-ring interaction network by structurally stabilizing the NTD, reinforcing the sequestration of ρ^R128 and the repositioning of ρ^K130 (Fig. 6b).

### Rof stabilizes the open ρ conformation and undercuts connector-mediated communication between ρ domains

Upon binding, Rof positions its β3-β4 loop on top of helix α5, establishing direct contacts of Rof^N48 to ρ^R128 and thus reinforcing the interaction of ρ^R128 with ρ^N25 of the neighboring NTD (Fig. 2g). Furthermore, Rof^K47 in the β3-β4 loop engages in a salt bridge with ρ^E24 of the neighboring ρ protomer (Fig. 2g). Thus, Rof bound between two neighboring NTDs of open ρ may stabilize their relative positions, and thus the open-ring conformation. To test whether Rof indeed counteracts ρ ring closure, we performed fluorescence anisotropy assays with the 5′-FAM-labeled SBS ligand, rU12, in the presence of unlabeled dC15 that blocks the PBSes[6], such that rU12 binding at the SBS reports on ρ ring closure (Fig. 5a). Indeed, while ρ alone readily underwent ring closure, Rof inhibited ring closure in a concentration-dependent manner (Fig. 5g).

Surprisingly, although ρ adopts an open conformation upon Rof binding, the connector retains the NTD/CTD-interaction register of the closed conformation (Fig. 6c, middle and bottom panels), presumably due to the direct Rof-helix α5 interactions restricting connector rearrangements. Concomitantly, in complex with Rof, the CTD Q-loops adopt a conformation observed in closed ρ, but not in open ρ structures in the absence of Rof[42] (Fig. 6d); in the closed ring as well as in the open ring in complex with Rof, ρ^K283 is inserted into a pocket formed by the Q-loop of the adjacent protomer, while ρ^K283 points to the center of the ring in open ρ in the absence of Rof (Fig. 6d).

Together, these observations show that Rof effectively stabilizes ρ in an open conformation by cross-strutting neighboring NTDs and preventing their separation, yet at the same time prevents a register shift and change in interactions along the NTD-CTD connector, maintaining the CTDs in a conformation resembling the closed state. Thus, Rof conformationally insulates ρ NTDs and CTDs by undercutting concerted structural transitions normally associated with ρ ring dynamics and induces a hybrid conformation, with the NTDs in the open state and the connector and CTDs in the closed-state conformation.

### Rof interferes with ρ binding to transcription elongation complexes

Recent structural studies had shown that ρ can engage ECs modified by general transcription factors NusA and NusG before binding the transcript, with the ρ NTDs establishing direct contacts to NusA, NusG and RNAP subunits α, β and β′[33,34]. In the ρ-bound NusA/NusG-ECs, ρ remains dynamic relative to the surface of RNAP, allowing ρ to position one of its PBSes next to the RNA exit channel to eventually capture the nascent transcript. Subsequent conformational changes in RNAP can trigger transcription inactivation.

Superposition of the ρ-Rof and ρ-EC structures revealed that ρ surfaces in contact with ECs encompass the extended PBSes, and that Rof binding to ρ is incompatible with ρ binding to an EC (Fig. 7a).

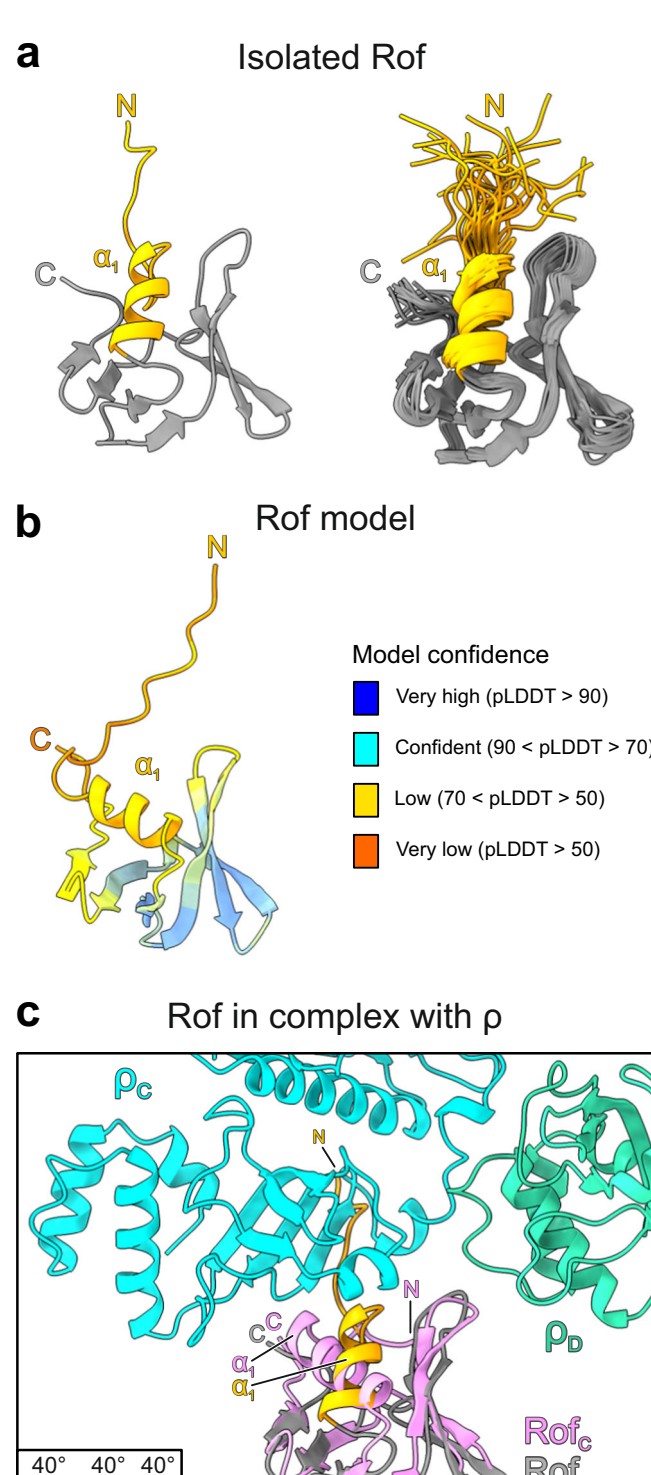

## a  Isolated Rof

## b  Rof model

**Model confidence**

- ■ Very high (pLDDT > 90)
- ■ Confident (90 < pLDDT > 70)
- ■ Low (70 < pLDDT > 50)
- ■ Very low (pLDDT > 50)

## c  Rof in complex with ρ

ρ_C  ρ_D

Rof_C
Rof_Isolated

40°  40°  40°

**Fig. 4 | Rof conformational changes upon ρ binding. a** Structure of *ec*Rof in isolation. Representative structure (left) and structural ensemble (right) of *ec*Rof determined by NMR spectroscopy (PDB ID: 1SG5)[21]. The N-termini (N) and helix α1 are colored in orange. C, C-termini. The N-termini of isolated *ec*Rof show high flexibility and are oriented away from the protein's core. **b** AlphaFold model of *ec*Rof. Rof is colored according to the AlphaFold model confidence score (pLDDT). The pLDDT score indicates structural flexibility in the N-terminal region, including helix α1. **c** Isolated *ec*Rof (gray) superimposed on ρ-bound *ec*Rof (violet) in the same view as isolated *ec*Rof in (**a**). Rotation symbols, view relative to Fig. 2b. In the conformation of isolated *ec*Rof, the N-terminus and helix α1 would sterically interfere with binding to ρ.

Consistently, when in complex with Rof, ρ showed impaired binding to a NusA/NusG-EC in analytical SEC (Fig. 7b; compare panels 2 and 3). Similarly, ρ dissociates from a pre-formed ρ/NusA/NusG-EC upon addition of Rof (Fig. 7b; compare panels 2 and 4), indicating that Rof can act on ρ-bound transcription complexes within the cell. Thus, in addition to preventing productive engagement of a transcript, Rof also hinders EC engagement by ρ.

### Rof proteins from *Vibrionaceae* may resort to a modified ρ-inhibitory mechanism

Unlike ρ, which is ubiquitous in bacteria, Rof is restricted to Pseudomonadota (synonym Proteobacteria; Fig. 1d); 79% of *Enterobacteriaceae* and 63% of *Vibrionaceae* genomes are estimated to encode Rof (Supplementary Data 1). In *E. coli*, *rof* is encoded in a five-gene locus bracketed by REP elements (Fig. 1e). Two genes in this cluster, *arfB* and *nlpE*, play vital roles in *E. coli* stress response. ArfB, a peptidyl-tRNA hydrolase, rescues stalled ribosomes when trans-translation control fails[43]. NlpE is an outer membrane lipoprotein that activates the Cpx pathway in response to envelope stress[44]. Rof is expressed from a divergent promoter in tandem with a small conserved protein of unknown function, YaeP[45].

Presence of *rof* in diverse Pseudomonadota (Fig. 1d) suggests a common mechanism of ρ inhibition. However, recombinant *V. cholerae* Rof (*vc*Rof) has been observed to form higher-order oligomers in vitro, of which the dimer state could be stabilized by treatment with iodoacetamide/TCEP; furthermore, *vc*Rof seems to disassemble ρ hexamers in vitro[22]. ρ-bound *ec*Rof and a *vc*Rof monomer (PDB ID: 6JIE)[22] exhibit similar structures that superimpose with a root-mean-square deviation (rmsd) of 1.05 Å for 58 pairs of Cα atoms (2.65 Å for all 72 pairs of Cα atoms). Similar to isolated *ec*Rof, isolated *vc*Rof differs from ρ-bound *ec*Rof in the orientation of the N-terminal region, which is diverted from the OB-fold; furthermore, helix α1 in isolated *vc*Rof is shortened by one turn compared to *ec*Rof (Fig. 8a, b). Given that crucial ρ-contacting residues in the N-terminal regions are highly conserved between *ec*Rof and *vc*Rof (Fig. 8c), *vc*Rof most likely binds its cognate ρ in a similar manner, undergoing conformational changes as revealed here for *ec*Rof. However, more pronounced sequence divergence in other regions of Rof proteins from *Enterobacteriaceae* and *Vibrionaceae* (Fig. 8c), including in the ρ-contacting β3-β4 loops (Fig. 2g), may reflect some differences in the specific contacts to ρ.

A heterotypic configuration of the *vc*Rof dimer has been proposed based on a crystal structure (PDB ID: 6JIE)[22]. While putative ρ-interacting surfaces, in particular the N-terminal region, of one *vc*Rof subunit are involved in the proposed dimerization, the putative ρ-interacting elements of the other *vc*Rof subunit are unobstructed; thus, while the proposed *vc*Rof dimer could engage ρ via one of the *vc*Rof subunits as observed for our *E. coli* ρ-Rof complexes without steric hindrance (Fig. 8d, top), conflicts would ensue upon dimeric *vc*Rof binding to ρ via the subunit with partially buried ρ-interacting elements (Fig. 8d, bottom). While such engagement may in principle lead to conformational changes in ρ and disassembly of the hexamer, the molecular mechanisms that would favor this disruptive engagement over binding of dimeric *vc*Rof via the unobstructed subunit remain unclear.

Apart from differences in the interaction details and the oligomeric states, other factors may differentially influence the mode of action of Rof proteins in *Enterobacteriaceae* and *Vibrionaceae*. We observed that the *rof* genomic contexts in *Enterobacteriaceae* and *Vibrionaceae* are strikingly different. While in *Enterobacteriaceae*, the *rof* gene always overlaps with an upstream *yaeP* gene, the *yaeP* and *rof* genes are located on different chromosomes in *V. cholerae* (Fig. 8e and Supplementary Data 1). Furthermore, *rof* does not have conserved gene neighbors in *Vibrionaceae* (Supplementary Fig. 10 and

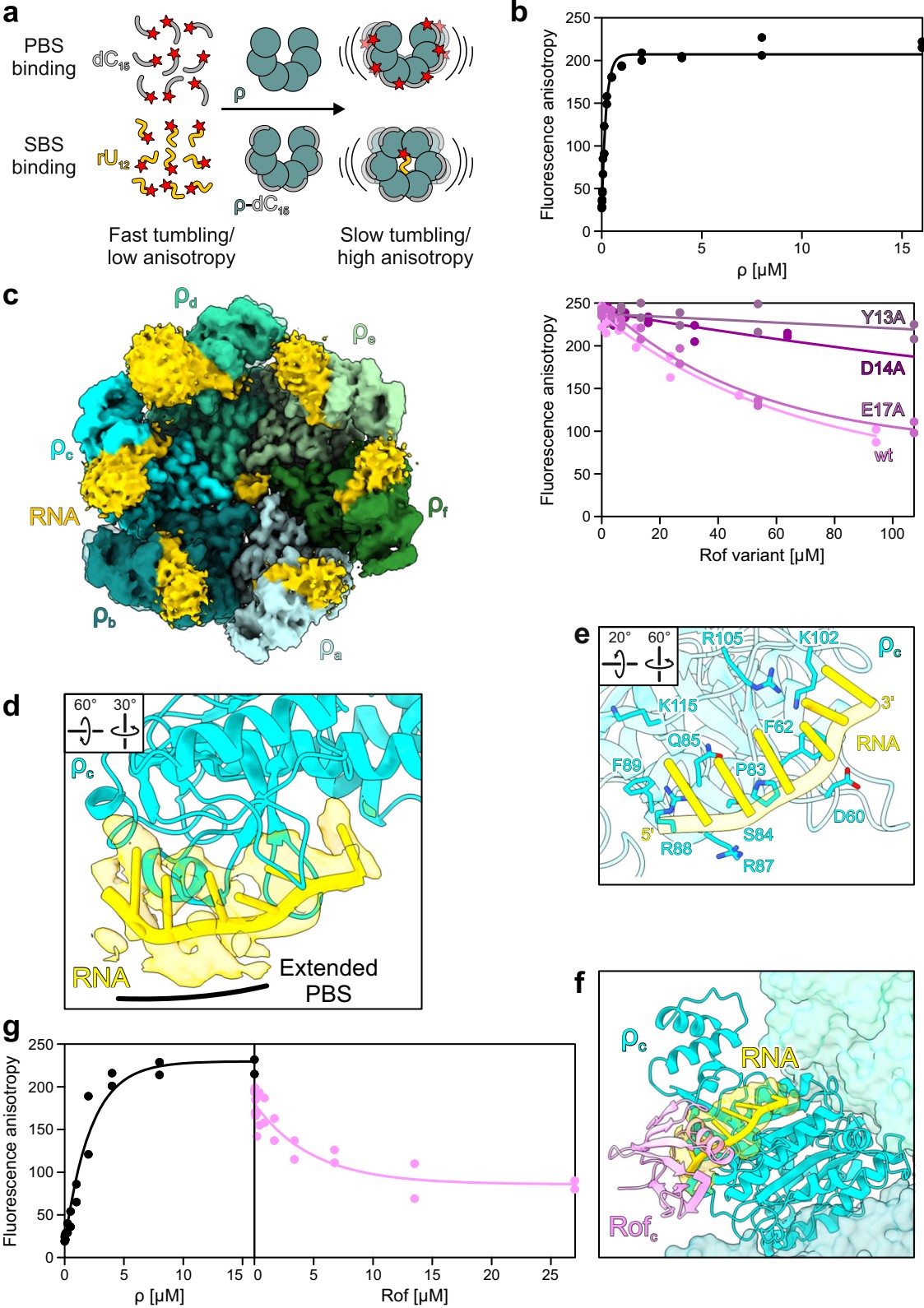

Supplementary Data 1). Collectively, our analyses reveal that monomeric *ec*Rof and dimeric *vc*Rof could engage ρ in an analogous fashion and thus inhibit ρ function via similar molecular principles. The molecular mechanisms of putative additional levels of ρ inhibition via subunit dissociation by *vc*Rof as well as possible differences in Rof regulation remain to be elucidated.

## Discussion

ρ is a global gene regulator and is essential in most bacterial species. It defines gene boundaries, silences production of useless and harmful RNA, and resolves R-loops[1]. Single-molecule fluorescence studies are consistent with the idea that ρ-dependent termination can proceed via two pathways[46,47]. In the classical RNA-dependent pathway, ρ binds

**Fig. 5 | Effect of Rof on RNA binding by ρ. a** Principles of fluorescence anisotropy assays used in (**b**) and (**g**). Isolated fluorescence-labeled (red star) $dC_{15}$ DNA oligos (gray) or $rU_{12}$ RNA oligos (gold) exhibit high tumbling rates (low fluorescence anisotropy). In the presence of ρ, $dC_{15}$ binds to the PBSes of ρ, which reduces its tumbling rate (increased fluorescence anisotropy). In the presence of $dC_{15}$, $rU_{12}$ binds to the SBS of ρ, which leads to ρ ring closure. **b** PBS binding. Top, binding of $dC_{15}$ to the ρ PBSes in the presence of increasing concentrations of ρ. Bottom, effect of Rof variants on the binding of $dC_{15}$ to the PBSes. Data of two independent experiments and fits to a single-exponential Hill function (see Methods) are shown. Source data are provided as a Source Data file. **c** CryoEM reconstruction of the ρ-*rut* RNA structure. ρ forms a closed ring and is bound by RNA at the NTD of each protomer and at the SBS. Only RNA regions bound at the SBS and at the $ρ_c$ NTD were modeled. RNA density and model, gold. As originally proposed[2,5], ρ subunits

are labeled in opposite directions around open and closed ρ rings (compare to Fig. 2b). We therefore labeled subunits of open ρ with capitals (A-F) and subunits of closed ρ with lower case letters (**a**–**f**). Rotation symbols in this and (**d**) views are relative to (**c**). **d** ρ interaction with RNA at the PBS. Six RNA residues (cartoon) fit to the density observed at $ρ_c$ (semitransparent surface). **e** Path of the RNA along the PBS. ρ residues are shown as sticks and labeled. **f** Structural comparison between ρ-*rut* RNA and $ρ_6$-ADP-$Rof_5$ complexes. Structures of the complexes were superimposed based on the $ρ_c/ρ_c$ subunits. Bound Rof sterically hinders RNA accommodation at the extended PBS. **g** Ring closure assays. Left, fluorescence anisotropy of 5′-FAM-labeled $rU_{12}$ in the presence of increasing amounts of ρ. Right, Rof inhibits ρ ring closure. Data of two independent experiments and fits to a single-exponential Hill function (see Methods) are shown. Source data are provided as a Source Data file.

C-rich, unstructured and unoccupied RNA regions via its PBSes; subsequent binding of a neighboring RNA region at the SBS and ring closure allow ρ to translocate towards elongating RNAP, eventually contact the transcriptional machinery around the RNA exit channel and terminate transcription[24,48]. In the EC-dependent pathway, which may also represent a prelude to ultimate termination via the classical mechanism[33], ρ associates with and accompanies ECs without immediately terminating transcription[49]; upon EC pausing, NusA and NusG can cooperate with ρ in inducing conformational changes in RNAP that stall transcription before ρ engages the transcript[33,34].

A deleterious mutation in *rho* enables *E. coli* survival during proteotoxic stress[50] and envelope stress turns on the expression of YihE, a Ser/Thr kinase that inhibits ρ/RNA interactions[20]. These findings suggest that ρ activity may need to be tuned down under some stress conditions. The RNA chaperone Hfq, which has been implicated in diverse stress responses[51], inhibits ρ function[16,19] as does another Sm-like protein, Rof[25], but their mechanisms remained incompletely understood. Here we present structural, functional and phylogenetic analyses of ρ regulation by *E. coli* Rof, which are in agreement with a study of *Salmonella* Rof by Zhang et al.[30].

Extending and detailing an earlier docking model[21], our results show that Rof undergoes pronounced conformational changes and folding/immobilization upon binding to the NTDs of an open ρ ring, thereby implementing multiple strategies to counteract both ρ-dependent termination pathways. First, Rof prevents ρ from binding to RNA at the PBS by blocking an extended PBS. An extended PBS has long been envisioned to contribute to both the RNA affinity and specificity of ρ, even though early structures of open ρ revealed only a dinucleotide bound at the core PBS (PDB ID: 1PVO)[2] and early structures of closed ρ lacked RNA at the PBS (PDB ID: 3ICE)[5]. A recent structure of a ρ pre-termination complex[24] disclosed RNA bound along extended PBSes and at the SBS of a closed ρ ring, as confirmed by our ρ-*rut* RNA structure. We show that ρ residues contacting Rof are also crucial for binding RNA at the extended PBS (Fig. 5d–f and Supplementary Fig. 7c) and are highly conserved (Supplementary Fig. 6b).

Second, Rof prevents ρ ring closure and thus RNA binding at the SBS (Fig. 5g). Given that optimal SBS ligands (e.g., $rU_{12}$) promote ring closure even in the absence of a PBS ligand[6], PBS blocking alone cannot fully accounts for Rof-mediated inhibition of ρ ring closure. Furthermore, as Rof itself binds at the ρ PBSes—why does Rof binding not induce ring closure? Our results show that Rof bridges the NTDs of two adjacent ρ protomers, thus preventing their separation that is required for ring closure (Fig. 2c, g).

Third, Rof conformationally insulates the ρ NTDs and CTDs, undercutting conformational dynamics involved in RNA binding at the SBS and ring closure. Functional studies had suggested that residues forming the core and extended PBSes are involved in an allosteric network that mediates communication across ρ domains via the NTD-CTD connector. For example, a Y80C substitution in the NTD alters the CTD conformation[40], and $ρ^{R88}$ has been suggested to affect steps other than RNA binding at the PBS[52]. Our ρ-*rut* RNA structure suggests that such

communication involves stabilization of the bound regions, which strengthens contacts of the NTD-CTD connector characteristic of the closed state. In addition, the 5′-end of PBS-bound RNA could directly contact the connector helix α5 and stabilize the closed-state configuration. Rof, in contrast, induces a chimeric structure in ρ wherein NTDs adopt an open-ring arrangement, while the NTD-CTD connector and the CTDs resemble the closed state, presumably due to Rof also directly contacting helix α5 of the connector, thus stabilizing the closed-state register (Fig. 6c). In particular, the CTD Q-loops adopt a conformation very similar to that observed in closed-ring structures of ρ, with $ρ^{K283}$ embedded in a pocket formed by the Q-loop of the neighboring protomer (Fig. 6d). In open ρ complexes in the absence of Rof, $ρ^{K283}$ side chains instead point to the ring interior (PDB ID: 6WA8)[42]. We speculate that $ρ^{K283}$ in open ρ might be important for the initial RNA capture at the SBS and its proper alignment during ring closure. Thus, by retaining CTDs in a state resembling the closed ring, and thereby sequestering $ρ^{K283}$, Rof might also counteract initial RNA engagement at the SBS.

Fourth, as the extended PBSes and other Rof-contacting surfaces on ρ also represent main contact points of ρ on NusA/NusG-modified ECs[33,34], Rof interferes with ρ binding to ECs and can partially dissociate ρ from pre-formed ρ-ECs (Fig. 7). As Rof binding at the ρ PBSes would, thus, equally affect both, the RNA-dependent and EC-dependent termination pathways, our results do not favor one over the other.

Rof-mediated inhibition of ρ complements other strategies of ρ-regulation in bacteria (Fig. 9). Maintaining a balance between inactive ρ sequestered by Rof and unobstructed ρ that could engage nascent transcripts or ECs would allow cells to tune ρ-dependent transcription termination in response to cellular cues, e.g., during slow growth or under stress. Consistently, we found that the absence of σ^S, a master regulator of general stress response, sensitizes *E. coli* to Rof expression (Supplementary Fig. 1b). At least two key questions concerning the cellular role of Rof presently remain unanswered: when does the cell need Rof and how is Rof expression controlled? Rof is present in many Pseudomonadota, implying that its ability to dampen ρ-dependent termination is important under some conditions. A recent study showed that while the *rof* deletion did not alter growth of *Salmonella* in standard laboratory conditions, it attenuated virulence by suppressing the expression of genes encoded on the SPI-I pathogenicity island[30].

As ρ is essential, a global anti-terminator should be produced in substantial amounts only when needed. Indeed, a quantitative proteomic analysis of *E. coli*[17] demonstrated that both Rof and YihE are present at low (and similar) levels across many conditions, with a modest increase during the stationary phase, consistent with the first report on Rof[25]. YihE is activated by Cpx[53], a two-component system that responds to envelope stress[44]. Collectively, it stands to reason that the expression of *yaeP-rof* is induced by particular cellular signals; e.g., a modest increase in the *yaeP-rof* transcript level was observed during recovery from exposure to mercury[54]. Identification of other triggers awaits a systematic analysis. Bacteria experience a wide variety of adverse conditions and inhibiting ρ could be essential for survival in some, but not other, situations. We hypothesize that Rof and YihE

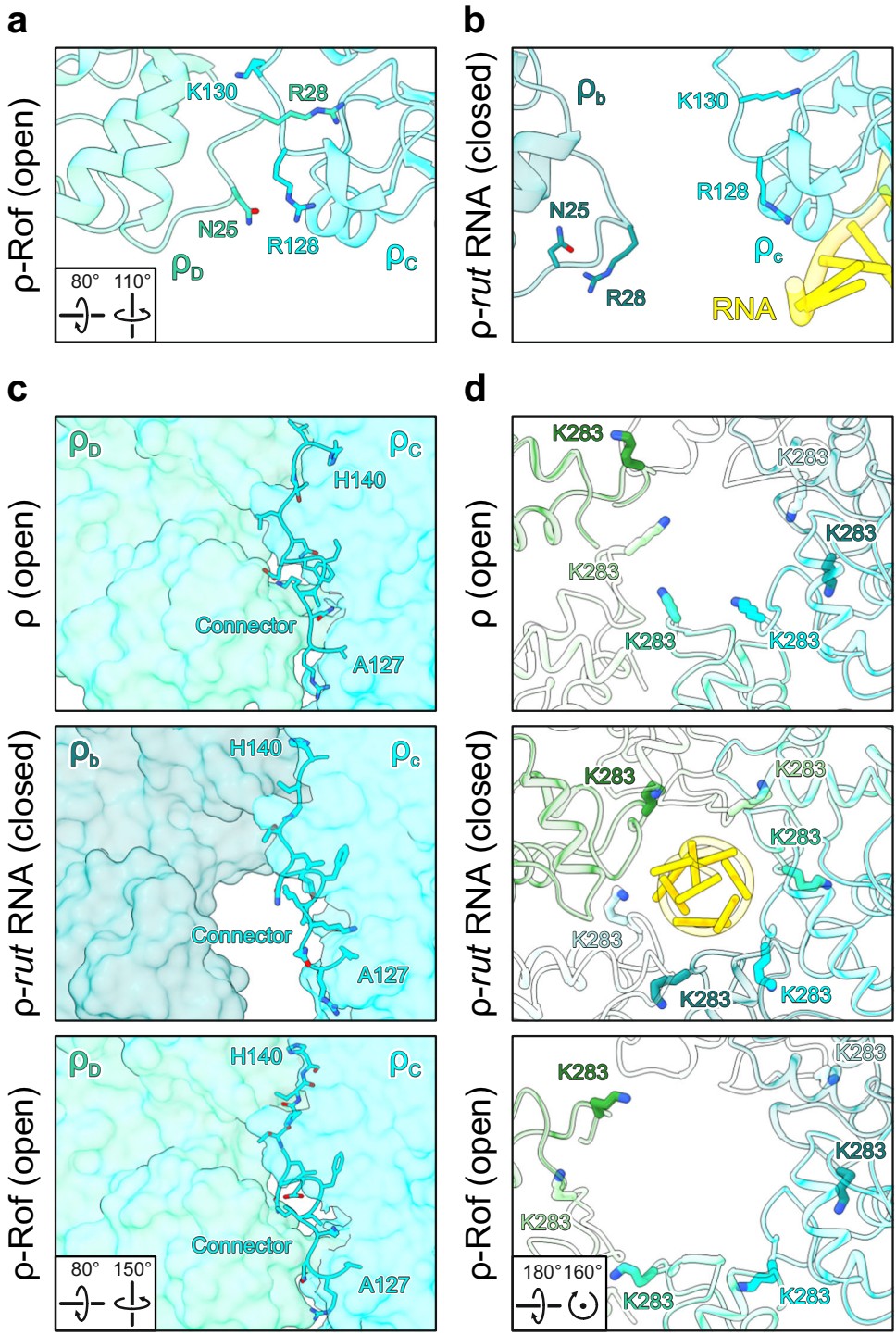

**Fig. 6 | Effects of Rof on ρ conformation. a, b** Structural comparison of ρ subunit interactions in structures of open ρ bound to ATP (PDB ID: 6WA8)[42] (**a**) and closed ρ bound to *rut* RNA (**b**). In the open hexamer conformation, residue R28 from one ρ protomer ($\rho_D$) is embedded in a pocket of the adjacent protomer ($\rho_C$), formed partially by $\alpha_5$. In the closed hexamer conformation, R28 is ejected from that pocket and substituted by K130 of the same protomer ($\rho_c$). Concomitantly, contacts between the N-terminal region of $\rho_D$ and R128 of $\rho_C$ are broken during ring closure ($\rho_b$ an $\rho_c$ in the closed hexamer). Rotation symbols, view relative to Fig. 5c. **c** Comparison of the NTD-CTD connectors (sticks) in open $\rho^{ATP}$ (top), the closed ρ-*rut* RNA complex (middle) and the open $\rho_6$-ADP-Rof$_5$ complex (bottom). Relative to open $\rho^{ATP}$, the NTD-CTD connector (residues 127–140) in the closed ρ-*rut* RNA

complex is re-aligned by one residue, most evident by the rearrangement of H140 towards the protomer interface. In the open $\rho_6$-ADP-Rof$_5$ structure, the NTD-CTD connector retains the register observed in the closed ρ-*rut* RNA complex. Rotation symbols in this and (**d**) views are relative to Fig. 2b. **d** Structural comparison of Q-loop conformations in open $\rho^{ATP}$ (top), the closed ρ-*rut* RNA complex (middle) and the open $\rho_6$-ADP-Rof$_5$ complex (bottom). In open $\rho^{ATP}$, Q-loops adopt a conformation resulting in residues K283 (sticks) pointing towards the central axis of the ρ hexamer, where they could engage in initial contacts to SBS RNA. In the closed ρ-*rut* RNA complex, K283 residues are embedded in pockets formed in part by the Q-loops of the adjacent ρ protomers. In the open $\rho_6$-ADP-Rof$_5$ complex, the Q-loops adopt a conformation similar to that observed in the closed ρ-*rut* RNA complex.

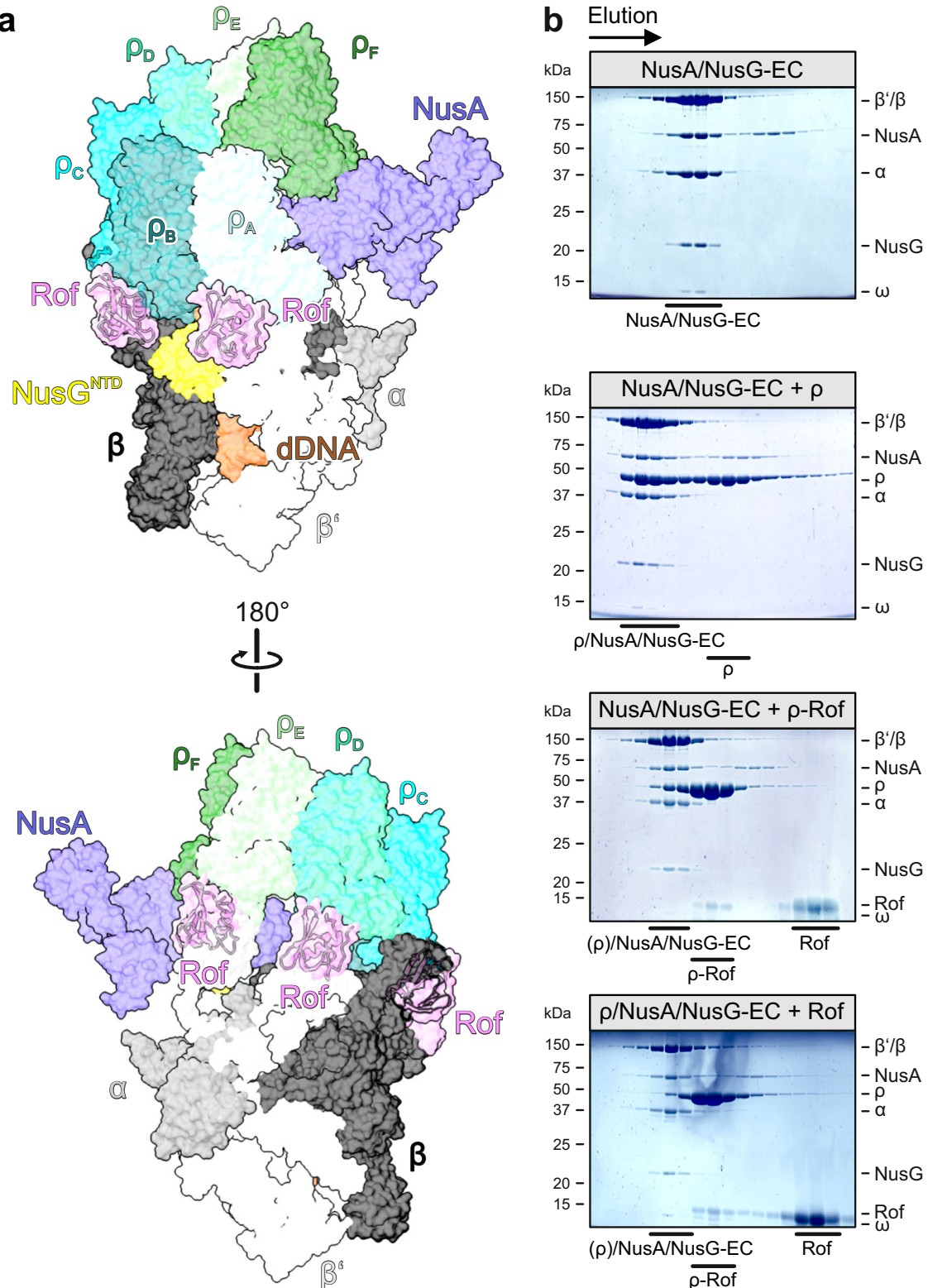

**Fig. 7 | Rof effects on ρ-EC interactions. a** Rof interferes with ρ binding to ECs. The ρ6-ADP-Rof5 structure was superimposed on the structure of a ρ/NusA/NusG-modified EC (PDB ID: 6Z9P)[33] based on the ρA protomers. RNAP subunits, different shades of gray; NusA, slate blue; NusG, yellow; downstream (d) DNA, brown. Rof sterically interferes with ρ binding to RNAP and Nus factors. **b** SDS-PAGE analysis of SEC runs, monitoring ρ binding to ECs in the absence and presence of Rof. First panel, NusA/NusG-EC. Second panel, pre-formed NusA/NusG-EC incubated with a three-fold molar excess of ρ hexamer. Third panel, pre-formed NusA/NusG-EC incubated with a three-fold molar excess (relative to ρ hexamer) of ρ-Rof complex. Fourth panel, pre-formed ρ/NusA/NusG-EC incubated with a ten-fold molar excess of Rof (relative to ρ hexamer). Experiments were performed three times independently with similar results. Source data are provided as a Source Data file.

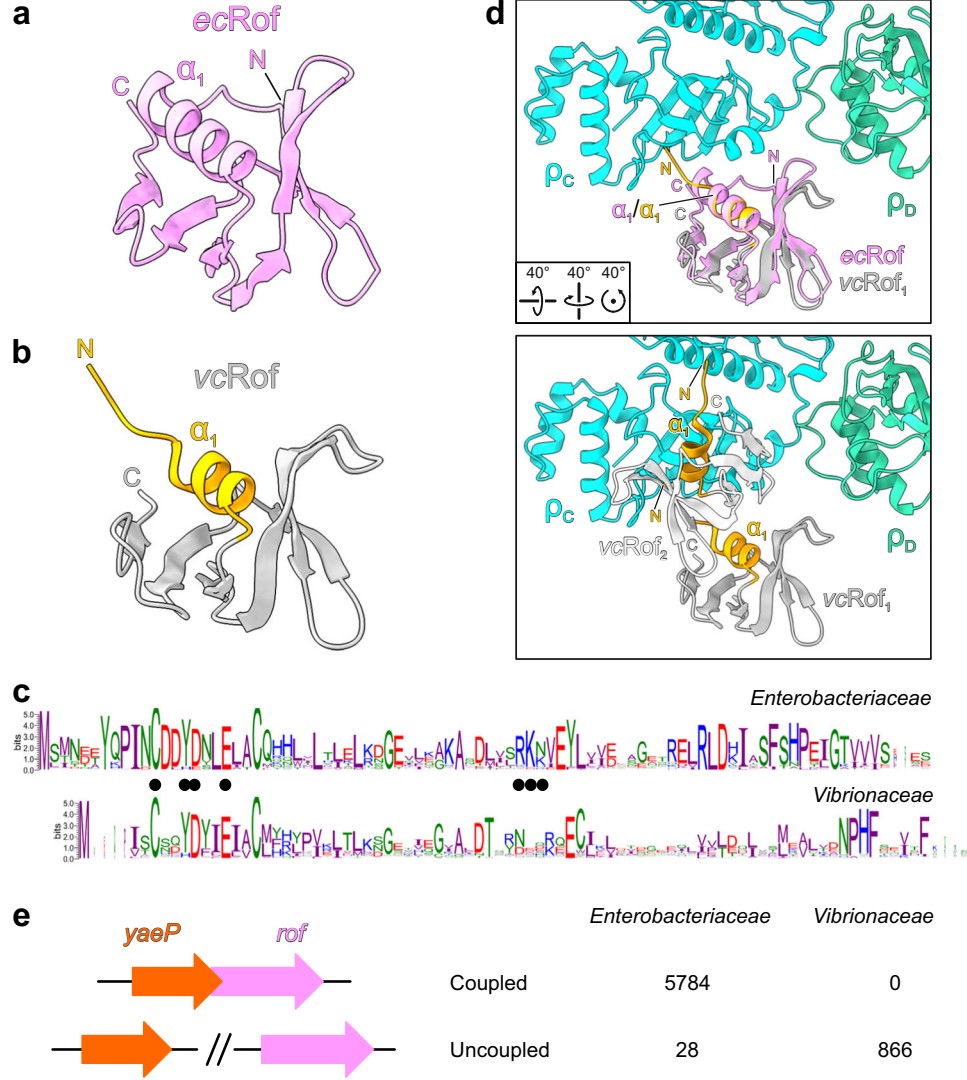

**Fig. 8 | Rof conformational changes and conservation. a, b** The structure of one *E. coli* (*ec*) Rof protein as observed in the ρ-Rof complexes (**a**) compared to one monomer from the crystal structure of a *V. cholerae* (*vc*) Rof dimer (**b**; PDB ID: 6JIE). *Ec*Rof bound to ρ and *vc*Rof mainly differ in the orientation of the N-termini and the lengths of helices α₁. N, N-termini; C, C-termini. **c** Conservation of Rof sequences in *Enterobacteriaceae* (top) and *Vibrionaceae* (bottom). In *Enterobacteriacea*, two alternative start sites for Rof are possible, generating 84- and 86-residue Rof. The interface residues revealed by the *E. coli* ρ-Rof complex structures are indicated by black dots. Sequence logos were generated by WebLogo (version 3.7.8). **d** Superposition of the *vc*Rof on *ec*Rof bound to ρ. Upper panel, superposition of a *vc*Rof monomer on *ec*Rof bound to ρ. Lower panel, superposition of the *vc*Rof dimer on *ec*Rof bound to ρ; for clarity, *ec*Rof is not shown. Rotation symbols, view relative to Fig. 2b. While monomeric *vc*Rof would align without steric conflict, there might be steric hindrance in the interaction between a dimer of *vc*Rof and ρ. **e** The *rof* and *yaeP* genes are coupled in *Enterobacteriaceae*, but not in *Vibrionaceae*; see Supplementary Data 1 for more details.

belong to a large set of specialized anti-termination factors, which could comprise proteins and sRNAs, that turn off ρ under diverse stress conditions to enable survival and facilitate recovery (Fig. 9).

## Methods

### Recombinant protein and RNA production and purification

Supplementary Table 3 lists plasmids used in this study. *E. coli* ρ protein and variants thereof were produced and purified as described before[11]. DNA encoding Rof was codon optimized for *E. coli* expression and purchased from Invitrogen (Thermo Fisher Scientific). The *rof* gene was cloned into pETM-11 and expressed in *E. coli* RIL (BL21) cells (Novagen). Cells were lysed by sonication in buffer A (100 mM KCl, 5 mM MgCl₂, 20 mM Na-HEPES, pH 7.5, 1 mM DTT, 10% [v/v] glycerol, 1 mM DTT) and the lysate was cleared by centrifugation at 55,914 x g for 1 h at 4 °C. The cleared lysate was loaded on a 5 ml Ni²⁺-NTA column (Cytiva) equilibrated in buffer A. Rof was eluted in elution buffer (buffer A supplemented with 500 mM imidazole). The eluate was treated with TEV

protease and dialysed overnight against buffer A with 5% (v/v) glycerol. After recycling on the Ni²⁺-NTA column, the flow-through was collected, concentrated and loaded to a Superdex 75 column (Cytiva) equilibrated in SEC buffer (100 mM KCl, 5 mM MgCl₂, 20 mM Na-HEPES, pH 7.5, 1 mM DTT). Peak fractions were pooled and concentrated to 4.4 mM. Rof was flash frozen in liquid nitrogen and stored at -80 °C until further use. *rho* and *rof* mutations were introduced by site directed mutagenesis using the QuikChange protocol (Stratagene) and protein variants were produced and purified as for the wt versions. In vitro transcribed *rut* RNA was produced and purified as described before[33].

### SEC-MALS

SEC-MALS was performed in reaction buffer (20 mM Tris-HCl, pH 8.0, 120 mM KOAc, 5 mM Mg(OAc)₂, 2 mM DTT). 1 mg/ml of ρ were mixed with 2 mg/ml of Rof and incubated at 32 °C for 10 min. 60 μl of the mixture were loaded on a Superose 200 increase 10/300 GL column (Cytiva) and chromatographed on HPLC system (Agilent) coupled to a

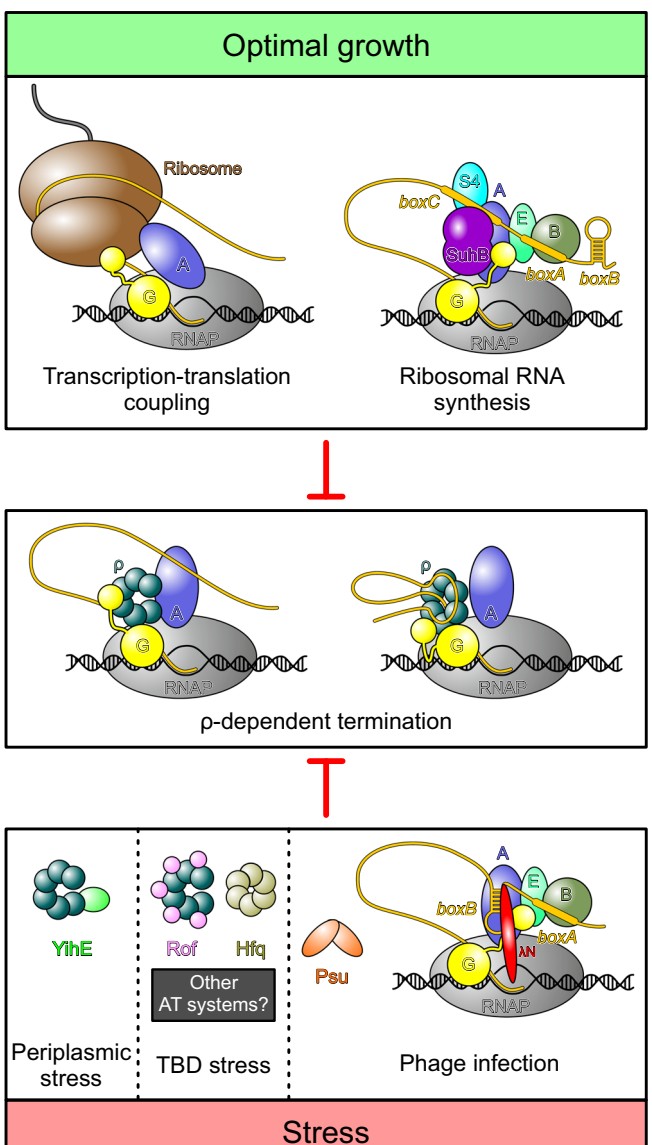

**Fig. 9 | ρ regulation under optimal growth and stress conditions.** Top, under optimal growth conditions bulk mRNA synthesis is protected from premature termination by ρ through transcription-translation coupling[66,67]. A specific transcription anti-termination complex (*rrn*TAC) shields the ribosomal RNAs from ρ[10]. Bottom, under stress conditions, when translation is inefficient, ρ activity must be regulated. Phages use different strategies to protect transcription of their own genomes. Similar to the *rrn*TAC, lambdoid phages rely on specific anti-termination complexes, such as λN-TAC, to shield the nascent transcript from ρ[11]. Phage P4 uses the Psu protein that directly binds to ρ and thereby prevents the formation of termination complexes[38]. To date, three cellular proteins are known to directly bind ρ and inhibit termination, but other stress-specific regulators likely exist. YihE is induced upon periplasmatic stress[44,53], whereas stress conditions under which Hfq and Rof regulate ρ activity remain to be identified. YihE binds the ρ NTD, inhibiting ρ-RNA interactions[20]; the mechanism of ρ regulation by Rof has been determined in this study, and molecular details of Hfq-mediated ρ inhibition remain to be solved[16,19]. By directly binding ρ, each of these anti-terminators may interfere with the formation of ρ-dependent termination complexes that assemble in an EC-dependent manner (center left)[33,34] or RNA-dependent manner (center right)[24,48].

miniDAWN TREOS multi-angle light scattering and a RefractoMax 520 detector system (Wyatt Technologies). Prior to measurements, a system calibration was performed using BSA (Sigma-Aldrich). ASTRA 6.1 software was used for data analysis and molecular mass determination.

## Sample preparation for cryoEM
For ρ-Rof complex formation, 41.8 μM ρ hexamer were mixed with a 10-fold molar excess of Rof in the absence or presence of 2.3 mM ADP-BeF$_3$ in reaction buffer and incubated for 15 min at 32 °C. The mixture was applied to a Superdex 200 Increase 3.2/300 column (Cytiva) and fractions of the complex were pooled and concentrated. 3.8 μl of the purified complex (5.3 mg/ml) were applied to glow-discharged Quantifoil R1.2/1.3 holey carbon grids and plunged into liquid ethane using a Vitrobot Mark IV (Thermo Fisher) set at 10 °C and 100% humidity.

For ρ-*rut* RNA complex formation, 41.8 μM ρ hexamer were mixed with an equimolar amount of *rut* RNA (Supplementary Table 3) and 2.3 mM ADP-BeF$_3$ in reaction buffer and incubated for 15 min at 32 °C. The mixture was applied to a Superose 6 Increase 3.2/300 column (Cytiva) and fractions of the complex were pooled and concentrated. Purified complex (3.8 mg/ml) was vitrified as above.

## CryoEM data acquisition and processing
CryoEM data were acquired on a FEI Titan Krios G3i TEM operated at 300 kV equipped with a Falcon 3EC direct electron detector (EPU, version 2.8.1; Thermo Fisher Scientific). Movies for ρ-Rof samples were taken for 40.57 s accumulating a total electron dose of ~ 40 e$^-$/Å$^2$ in counting mode distributed over 33 fractions at a nominal magnification of 96,000 x, yielding a calibrated pixel size of 0.832 Å/px. Data for the ρ-*rut* RNA sample were acquired at a higher nominal magnification of 120,000 x, corresponding to a calibrated pixel size of 0.657 Å/px. A total electron dose of 40 e$^-$/Å$^2$ was accumulated over an exposure time of 30.58 s.

All image analysis steps were done with cryoSPARC (version 4.1.1)[55]. Movie alignment was done with patch motion correction, CTF estimation was conducted with Patch CTF. Class averages of manually selected particle images were used to generate an initial template for reference-based particle picking from 4216 micrographs for the ρ-Rof sample. 1,782,459 particle images were extracted with a box size of 384 px and Fourier-cropped to 96 px for initial analysis. Reference-free 2D classification was used to select 1,101,769 particle images for further analysis. Ab initio reconstruction was conducted to generate an initial 3D reference for heterogeneous 3D refinement. 600,888 particle images were further classified by 3D variability analysis into ρ$_6$-Rof$_5$ and ρ$_5$-Rof$_4$ complexes. Particle images were re-extracted with a box size of 384 px after local motion correction and subjected to non-uniform refinement giving reconstructions at ~ 2.9 Å resolution each that could be slightly improved by CTF refinement. Another iteration of heterogeneous 3D refinement was applied to determine the final set of 110,264 particle images for the ρ$_6$-Rof$_5$ complex and 293,352 particle images for the ρ$_5$-Rof$_4$ complex. Non-uniform (NU) refinement yielded reconstructions at a global resolution of 2.92 Å and 2.74 Å, respectively. Data analysis for the ρ-Rof samples was conducted similarly.

## Model building, refinement and analysis
Coordinates of ρ (PDB ID: 6WA8 [open ring]; PDB ID: 1PVO [closed ring]) and Rof (PDB ID: 1SG5) were docked into the cryoEM maps using Coot (version 0.8.9)[56]. ρ subunits and Rof were manually adjusted to fit the cryoEM density. The *rut* RNA regions were built de novo except for the dinucleotide at the core PBS (taken from PDB ID: 1PVO)[2]. Manual model building alternated with real space refinement in PHENIX (version 1.20.1)[57]. Data collection and refinement statistics are provided in Supplementary Table 2. Structure figures were prepared with ChimeraX (version 1.6.1)[58].

## Analytical size-exclusion chromatography
Proteins and/or nucleic acids were mixed in reaction buffer and incubated at 32 °C for 10 min. 50 μl of the samples were loaded on a Superdex S200 P.C 3.2 column (ρ-Rof) or Superose 6 Increase 3.2/300 column (ρ-*rut* RNA) and chromatographed at a flow rate of 50 μl/min at 4 °C. To test for ρ-Rof interaction, 5 μM ρ and 90 μM Rof/variants were

used. To test for ρ-*rut* RNA interaction, 4 μM ρ/variants and 5 μM *rut* RNA were used.

For testing ρ binding to ECs in the absence or presence of Rof, ECs composed of RNAP, NusA, NusG, nucleic acid scaffold containing a short *rut*-less RNA (16 nts) without or with ρ were formed as described before[33]. 2.4 μM EC lacking ρ were incubated with a 3-fold molar excess of ρ-Rof complex at 32 °C for 15 min. Alternatively, 1.1 μM ρ-EC were incubated with a 10-fold molar excess of Rof at 32 °C for 15 min. Fractions were analyzed by SDS-PAGE and urea PAGE to reveal protein and nucleic acid contents, respectively. DNAs and RNAs used for complex assembly are listed in Supplementary Table 3.

### Nucleic acid binding assays

Nucleic acid binding to ρ PBS or SBS was tested by fluorescence depolarization-based assays[6]. For PBS binding, 5 μM 5′-FAM-labeled $dC_{15}$ oligo (Eurofins) were mixed with increasing amounts of ρ (0 to 17 μM final hexamer concentration) in 20 mM Na-HEPES, pH 7.5, 150 mM KCl, 5% (v/v) glycerol, 5 mM $MgCl_2$, 0.5 mM TCEP. For SBS binding, ρ PBSes were first saturated with 10 μM non-labeled $dC_{15}$. ρ (0 to 16 μM final hexamer concentration) was then mixed with 2 mM ADP-$BeF_3$ and 2 μM 5′-FAM-labeled $rU_{12}$ oligo (Eurofins). To test the effect of Rof and Rof variants on nucleic acid binding to ρ PBS or SBS, 1 or 5 μM of ρ hexamer were mixed with increasing amounts of Rof (0 – 100 μM) prior to $dC_{15}$ addition. The fluorescence anisotropy was recorded in OptiPlateTM 384-well plates (PerkinElmer) using a Spark Multimode Microplate reader (Tecan; excitation wavelength, 485 nm; detected emission wavelength, 530 nm). Two technical replicates were averaged for each sample and the data were analyzed with Prism software (version 9.0.2; GraphPad). To quantify ρ PBS or SBS binding, data were fitted to a single exponential Hill function; $Y = Y_{max}[\text{protein}]^h / (K_d^h + [\text{protein}]^h)$; $Y_{max}$, fitted maximum of nucleic acid bound; $K_d$, dissociation constant; h, Hill coefficient.

### Plating efficiency assays

Individual colonies of each strain, in 3 biological replicates, transformed with plasmids carrying inserts of interest under the control of $P_{trc}$ promoter were grown overnight in LB supplemented with carbenicillin at 32 °C. Serial dilutions of overnight cultures were spotted onto LB plates containing carbenicillin with and without IPTG.

### Growth assays

Individual colonies of each strain, in 4 biological replicates, were inoculated in MOPS EZ rich defined media (Teknova, #M2105) supplemented with carbenicillin (100 mg/l) and grown overnight at 32 °C. The overnight culture was diluted (1.5 μl into 98.5 μl) in fresh media supplemented with 1 mM IPTG, loaded into a 96-well plate, and grown in a BioTek EPOCH2 microplate reader at 32 °C under control of Gen5 software (version 3.12; BioTek). The $OD_{600}$ in each well was measured in 15 min intervals for 24 h with kinetic shaking and incubation. Results were analyzed in Microsoft Excel.

### Western blotting

Overnight cultures of cells transformed with plasmids encoding Rof variants were grown in LB + carbenicillin (100 mg/l) at 32 °C. After 1:100 dilution into fresh media, the cultures were grown to an early exponential phase at 32 °C, induced with 1 mM IPTG for 60 min, pelleted and frozen. Cell pellets were resuspended in PBS, sonicated and centrifuged. Protein concentration in cleared lysates was measured with Bradford reagent and normalized before resolving in SurePAGE Bis-Tris 8-16% gels (GenScript). Molecular weight markers were run on every gel to monitor separation. Proteins were transferred to a nitrocellulose membrane (Bio-Rad Trans-Blot Transfer Medium, Cat. 162-0112). Membrane was incubated with primary anti-HA antibodies from mice (1:5,000 dilution; Sigma-Aldrich, Cat.

H9658) and with secondary anti-mouse horseradish peroxidase linked antibodies (1:10,000 dilution; Amersham, Cat. NA931V) before imaging using Bio-Rad Clarity Max Western ECL Substrate (Cat. 1705062 S) with a Bio-Rad ChemiDoc XRS+ system controlled using ImageLab (version 6.1; Bio-Rad).

### In vitro termination assays

RNAP and ρ for in vitro assays were purified as described before[33]. RNAP holoenzymes were assembled by mixing the core RNAP with a three-fold molar excess of $\sigma^{70}$ transcription initiation factor, followed by incubation at 30 °C for 20 min. DNA templates were generated by PCR amplification with primers listed in Supplementary Table 3 and purified by PCR cleanup kit (Qiagen). Halted A26 ECs were formed at 37 °C for 12 min by mixing 50 nM RNAP holoenzyme with 25 nM DNA template, 100 μM ApU, 10 μM ATP and UTP, 2 μM GTP and 5 Ci/mmol [α$^{32}$P] GTP in ρ termination buffer (40 mM Tris-Cl, 50 mM KCl, 5 mM $MgCl_2$, 0.1 mM DTT, 3% (v/v) glycerol, pH 7.9). After addition of ρ (25 nM) and Rof variants (5 μM), the reactions were incubated for 3 min at 37 °C. Transcription was restarted by addition of a pre-warmed (at 37 °C) mixture containing 150 μM NTPs and 50 μg/ml rifapentin and incubated at 37 °C for 7 min. Reactions were quenched by addition of an equal volume of stop buffer (45 mM Tris-borate, 10 M urea, 20 mM EDTA, 0.2% xylene cyanol, 0.2% bromophenol blue, pH 8.3) and separated by denaturing 5% PAGE (7 M urea, 0.5X TBE). Gels were dried and products were visualized using a Typhoon FLA 9000 PhosphorImaging system (GE Healthcare). Readthrough and termination RNA products were quantitated with ImageQuant and Microsoft Excel software.

### Rof and ρ distribution in Pseudomonadota

Rof and ρ distribution (Supplementary Data 1) were estimated using Rof Pfam (PF07073) and ρ TIGR (TIGR00767) models in Annotree[59] with an e-value 10$^{-5}$. The phylogenetic tree was downloaded from Annotree. The result was visualized in R (version 4.2.2) with ggtree package[60].

### Sequence conservation and genomic context analysis

To investigate the conservation of Rof and ρ in Pseudomonadota, the GTDB[61] bacterial taxonomy list (r_202) was downloaded. Only complete genomes were used to build the database. Three representatives from each Genus of Pseudomonadota were selected randomly. Rof was identified with the Pfam model. To identify ρ, Pfam model Rho_RNA_bind (PF07497) was searched against the representative database using hmmsearch (version 3.3)[62]. The identified ρ-like proteins were further confirmed by searching NCBI HMM model (NF006886.1) against them using hmmsearch[62] (Supplementary Data 1).

To further investigate the conservation of Rof in *Enterobacteriaceae* and *Vibrionaceae*, reference genomes of *Enterobacteriaceae*, *Enterobacteriaceae*_A and *Vibrionaceae* were selected from the GTDB bacterial taxonomy list. One representative from each strain was used to build the database. Again, Rof was identified by its Pfam model (Supplementary Data 1).

Before performing multiple sequence alignment, sequence duplicates were removed using cd-hit (version 4.8.1)[63]. Sequences were aligned using MUSCLE (version 5.1)[64] with default settings. Sequence logos were generated using WebLogo (version 3.7.8)[65].

*Enterobacteriaceae* and *Vibrionaceae* genomes containing *rof* and *yaeP* were analyzed to determine if their ORFs overlap. The overlaps were determined based on chromosomal location, strand and gene distance. We note that instances of non-overlapping genes could be an artifact of low quantality genome assemblies.

### Reporting summary

Further information on research design is available in the Nature Portfolio Reporting Summary linked to this article.

## Data availability

CryoEM reconstructions have been deposited in the Electron Microscopy Data Bank (https://www.ebi.ac.uk/pdbe/emdb) under accession codes EMD-17875 ($\rho_6$-Rof$_5$), EMD-17877 ($\rho_5$-Rof$_4$), EMD-17874 ($\rho_6$-ADP-Rof$_5$), EMD-17876 ($\rho_5$-ADP-Rof$_4$) and EMD-17870 ($\rho$-*rut* RNA). Structure coordinates have been deposited in the RCSB Protein Data Bank (https://www.rcsb.org) with accession codes 8PTN ($\rho_6$-Rof$_5$), 8PTP ($\rho_5$-Rof$_4$), 8PTM ($\rho_6$-ADP-Rof$_5$), 8PTO ($\rho_5$-ADP-Rof$_4$) and 8PTG ($\rho$-*rut* RNA). Uncropped gel images and quantifications are provided in a Source Data file. All other data are contained in the manuscript or the Supplementary Information files. Structure coordinates used in this study are available from the RCSB Protein Data Bank [https://www.rcsb.org] under accession codes 1PVO, 1SG5, 3ICE, 5JJI, 6JIE, 6WA8 and 6Z9P. Source data are provided with this paper.

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

## Acknowledgements

We are very grateful to P. Lydia Freddolino and Natacha Ruiz for many stimulating discussions. This work was supported by the National Institutes of Health (GM067153 to I.A.), the Deutsche Forschungsgemeinschaft (INST 130/1064-1 FUGG to Freie Universität Berlin; GRK 2473 "Bioactive Peptides", project number 392923329, to M.C.W.; WA 1126/11-1, project number 433623608, to M.C.W.) and the Berlin University Alliance (501_BIS-CryoFac to M.C.W.). The Sanger Sequencing at the OSU Comprehensive Cancer Center is supported in part by NCI P30 CA0168058.

## Author contributions

N.S., B.W., T.L.S., D.G. and I.A. performed preparative and analytical biochemical experiments. M.F. and I.A. conducted in vivo experiments. B.W. carried out bioinformatic analyses. N.S. and T.H. prepared samples for cryoEM. T.H. collected cryoEM data and performed SPA. N.S. built and refined atomic models. All authors contributed to the analysis of data and the interpretation of the results. N.S. and I.A. wrote the manuscript with contributions from all authors. I.A. and M.C.W. coordinated the project.

## Funding

## Competing interests

The authors declare no competing interests.
