## [Peer Review File · Nature Communications]

Sm-like protein Rof inhibits transcription termination factor ρ by binding site obstruction and conformational insulationREVIEWER COMMENTS

Reviewer #1 (Remarks to the Author):

Said et al examine the mechanism of action of the E coli Rof protein, an inhibitor the Rho transcription terminator, by cro-EM, biochemical and in vivo assays. The various approaches are consistent: Rof prevents closure of the Rho hexamer ring. Along the way, they establish the existence of a previously hypothesized extended PBS, which contacts Rho residues. The work is thorough and convincing but needs clarification of the points listed below

1.To determine at what stage of cellular growth Rof expression is inhibitory, we monitored the growth kinetics of IA227". The following paragraph is not clear.

What the authors are looking at are the effects of the kil gene of the cryptic prophage and not directly at Rop. And what are the survivors Rop- or Kil resistant?

2. From the Discussion " Under stress conditions when dedicated anti-termination machineries and translating ^[1-]_[SEP] ribosomes are scarce, p action may need to be tuned down "

Why? With fewer ribosomes, one would want less mRNA, not more. Also, where is the evidence that Rop is more active or more essential during stress?

3. "the rof deletion in E. coli has no known growth phenotypes ^[1-]_[SEP]"

Which phenotypes – response to stress? Does overproduction of Rof slow the growth of stressed E. coli?

4. The Discussion details two competing models of Rho-mediated termination. Does the present work favor one of these? ^[1-]_[SEP]

5. Are Rof deletions synthetic lethal to YihE ^[1-]_[SEP] deletions?

6. "Rof itself is a PBS ligand" ^[1-]_[SEP]

Is this incompatible with the statement : "bound Rof ^[1-]_[SEP] would block RNA binding at the extended PBS (Fig. 5e)"? Why then would it have to be bound?

7. Figure 9 (Model) is not instructive; there is no panel showing Rof bound to anything.

Reviewer #2 (Remarks to the Author):

The authors report a cryo-EM structure of transcription termination factor Rho in complex with the transcription antitermination factor Rof. The structure shows that Rof inhibits Rho function by occluding the Rho primary binding site (PBS) for RNA and by interfering with formation of the catalytically competent, closed-ring conformational state of Rho. The manuscript will be of interest to researchers in bacterial transcription and transcriptional regulation and should be acceptable for publication in Nature Communications after revision to credit published results and to differentiate between new and confirmatory results.

Specific comments:

lines 79-82: Delete sentences starting with "To further..." and "This structure revealed." The results described in these two sentences confirm results reported in ref. 47. They do not "reveal" anything.

lines 243-289: Shorten this section to one short paragraph. (Results in this section are strictly confirmatory.) Start the paragraph with a sentence stating that ref. 47 reported structures of Rho pre-termination complexes that defined the full interaction between Rho PBS and RNA., and continue with a sentence stating that the authors obtained a structure that confirms the interactions between the Rho PBS and rut RNA reported in ref 47.

line 296: Start paragraph with sentence stating "The classical [cite reviews] and recently confirmed [cite ref. 47] mechanism of Rho-dependent termination involves ATP-hydrolysis-dependent 5'-to-3' translocation of Rho on RNA."

lines 314-316: Delete sentence starting with "However"; replace "in our Rho-rut structure" by "in the structure of the Rho pre-termination complex of ref. 47 and in our Rho-rut structure"; and replace "solve this riddle" by "explain this phenomenon."

lines 320-325: Delete.

line 417: Replace "In vitro" by "It has been hypothesized that."

line 419: Insert "classical" before "RNA-dependent."

line 420: Insert "hypothesized" before "EC-dependent."

lines 439-441: Replace "in a manner similar to our Rho-rut structure" by "as confirmed by our Rho-rut structure," and delete "; however...(Fig 5d)."

Reviewer #3 (Remarks to the Author):

Said et al reported the cryo-EM structure of ρ /Rof and ρ /rut RNA complexes describing the antitermination mechanism driven by Rof which has not been well characterized before. Rof is an antitermination factor and binds to the PBses site and blocks PBS/RNA interactions, preventing ρ from terminating the transcription. Moreover, the authors suggest that during the EC-dependent termination events, Rof also competes with NusA/NusG/EC complex preventing the assembly of pre-termination ρ /NusA/NusG/EC complex as an antitermination mechanism. This well-rounded article describes the structural details of ρ /Rof, and ρ /rut RNA interactions and proposes the Rof-dependent mechanisms of antitermination. The authors support the proposed mechanism with solid biochemical and genetic pieces of evidence and also imply that Rof-driven antitermination may have differences within Vibrionaceae in comparison to Enterobacteriaceae. There are several concerns, especially those on the cryo-EM maps and coordinates that need to be addressed.

Major Concerns:

1. About the cryo-EM maps and coordinates.

(1) Rho5-ADP-Rof4: The densities of 5 copies of Rho are good, but those of Rof are poor. 4 ADP molecules were built. Three ones have density while the one on chain E has very poor density, which should be removed from the model. The densities of Rof on the Rho-Rof interfaces are poor (almost no side chain densities, $\sim 6\text{\AA}$). Those need to be clarified in the text. The quality of the density doesn't look like the claimed 2.7Å. A further local refinement on Rof may be useful for improving the density.

(2) Rho6-ADP-Rof5, Rho5-Rof4, and Rho6-Rof5: Same as what has been observed in Rho5-ADP-Rof4 structure: good density of Rho, local resolution of Rof is low ($\sim 5\text{-}6\text{\AA}$).

(3) Rho-rutRNA: very good density and models in most parts. Those of ADP are clear. The part of RNA bound in the center of hexameric Rho is clear, but the claimed part bound around Chain C Tyr80 (PBS) is relatively poor, it's impossible to build more than 2nt RNA residues. In addition, the RNA sequence in the Rho center (SBS) might refer to the previous structure (the author didn't clearly state how to build RNA in the methods), but it's impossible to determine what sequence should be placed around chain C Tyr80 (PBS). Authors should remove the unreliable RNA part and provide additional evidence to support how and why they build such RNA sequences here.

(4) Figures of model-to-map FSCs need to be present to demonstrate the quality of the models and the corresponding ones (model resolution when FSC=0.5) should also be included in Table S2. The accuracy of the model is important for structural explanations and descriptions.

(5) Additional figures showing the density of different parts of the complex should be included to justify the claimed 3.3-2.7 resolutions.

(6) It is hard to understand the reported resolutions in the "Resolution (A)" part in Table S2. Please follow the general format shown in other cryo-EM papers.

2. In the text, authors described and showed the main interaction on the Rho-Rof interface, which is involved in 7 residues of Rof, but they only performed detailed mutagenesis and functional studies on part of them. In some places of the text, they only emphasized three, but also discussed other residues in other places. All these make reading confusing. The authors should clarify the logic of why and how they chose residues to do the mutagenesis and functional studies.

3. Further kinetic studies may be necessary to confirm the competition between EC/NusA/NusG/ ρ and EC/NusA/NusG/ ρ +Rof suggested by Figure 7 and support the claim that Rof also competes with NusA/NusG/EC complex preventing the assembly of pre-termination ρ /NusA/NusG/EC complex.

Minors:

1. lines 320-325 could be an overstatement unless confirmed by improving density.

"It is also conceivable, albeit not resolved in our ρ -rut RNA 320 structure, that the 5'-end of RNA regions bound at the extended PBS directly contacts the 321 neighboring connector helix $\alpha 5$. PBS-bound RNA could thereby stabilize the connector in the 322 closed-ring configuration and support an apparent rotational movement of the NTD, which 323 facilitates displacement of the three-helix bundle from the neighboring NTD, thus supporting 324 retraction of ρ R28 from its inter-subunit binding pocket (Fig. 6a,b)."

May be moved to the discussion section instead as a logical assumption.

2. Figure 8D may need additional modification (clearer label or ecRof and maybe an additional panel with actual superposition of the monomers (not dimer) to confirm the statement in the main text).

3. A new panel of superimposition in Figure 4 is necessary to show what conformational change is observed upon Rho binding.

Reviewer #4 (Remarks to the Author):

In the manuscript titled "Sm-like protein Rof Inhibits Transcription Factor ρ by Binding Site Obstruction and Conformational Insulation," Said et al. demonstrated the molecular mechanism of anti-termination by Rof using cryo-EM structures of the Rof-bound ρ complex. *E. coli* Rof (or YaeO) is a 9-kDa acidic protein whose expression is regulated by the growth rate. It stably binds ρ and reduces ρ -dependent termination. While its anti-termination activity in vivo was known, its in vitro activity was not confirmed. To gain a molecular understanding of how Rof inhibits ρ , the authors initially found that N-terminal histidine tagging negates Rof's activity by examining the effects of Rof variants on cell growth. Subsequently, the authors demonstrated that tag-cleaved Rof indeed inhibits ρ -dependent termination using an in vitro transcription assay and solved the cryo-EM structure of the ρ -Rof complex. In order to compare the Rof-bound ρ structure with a native structure, the team also solved the ρ -rut RNA complex structure. Although pre-termination complex structures containing ρ , rut RNA, NusG, NusA, and RNAP elongation complexes are known and were published by the same group, this ρ -RNA structure was crucial for a precise analysis of Rof's effect on ρ .

In the ρ -Rof complex structure, the authors discovered that the ρ hexamer of an open state is bound to four or five Rof molecules. Each ρ protomer binds one Rof, though the ρ protomers at the opening displayed relatively weak or no Rof density. The authors demonstrated that a single mutation of the Rof residues responsible for hydrogen bonds or salt bridges at the ρ -Rof interface results in the disruption of ρ -Rof complex formation using analytical SEC, in vitro transcription assay, and growth rate measurement. Rof exhibited significant conformational changes upon ρ binding when compared to its previously resolved crystal structure.

Next, the authors illustrated the competition between Rof and rutRNA for binding to ρ through the use of the ρ -RNA complex structure and fluorescence anisotropy. Furthermore, they revealed that Rof-bound ρ adopts an open state, while its NTD-CTD connector region resembles that of a closed ρ . This observation suggests that Rof not only competes with substrate RNA, particularly the PBS (primary RNA-binding site) substrate but also induces conformational changes in ρ , disrupting the communication between NTD and CTD. This communication is crucial for ρ 's ability to modify the RNAP elongation complex leading to transcription termination.

In conclusion, this review strongly recommends the publication of this manuscript in *Nature Communications*. Below, you'll find some additional comments on the manuscript.

Minor points

1. In Fig 6a, the ρ D and ρ C were drawn whereas ρ B and ρ C were were in Fig 6b. Was the choice of protomers with different subscripts intentional? In the figure legend, this reviewer is unsure about the comment in the parentheses, specifically "(ρ B an ρ C in the closed hexamer)".

2. In lines 267-273, although the manuscript describing that the boxB structure was used as a reference point for the RNA building, this reviewer still cannot understand how the 6-nt portion of rutA bound at the PBS of pc could be built. Was there a continuous RNA density observed between the boxB patch and the 6-nt If so, RNA? Could you provide a confident measurement of the number of bases positioned between the boxB patch and the 6-nt RNA?

Response to Reviewer Comments

Reviewer comments are repeated in bold, responses are in regular font, changed text passages are highlighted in yellow. Line numbers quoted in the responses refer to line numbers in the combined revised manuscript and SI file with marked changes.

Coordinate and map files as well as PDB validation reports are available at: <https://box.fu-berlin.de/s/jj3AFH8Ea4STqGz>

Reviewer #1 (Remarks to the Author):

Said et al examine the mechanism of action of the E coli Rof protein, an inhibitor the Rho transcription terminator, by cro-EM, biochemical and in vivo assays. The various approaches are consistent: Rof prevents closure of the Rho hexamer ring. Along the way, they establish the existence of a previously hypothesized extended PBS, which contacts Rho residues. The work is thorough and convincing but needs clarification of the points listed below.

We thank the reviewer for considering our work thorough and convincing.

- 1. To determine at what stage of cellular growth Rof expression is inhibitory, we monitored the growth kinetics of IA227". The following paragraph is not clear. What the authors are looking at are the effects of the kil gene of the cryptic prophage and not directly at Rop.**

The resistance to Rof cannot be solely attributed to the lack of prophages. Rof and BCM have been reported to have similar effects on regulation of small RNAs (Morita et al., 2022; reference 23 in the revised manuscript). This prompted us to test the effect of Rof overexpression in MDS42, which is resistant to BCM and, as we showed, to Rof. A natural conclusion may be that resident prophages, most likely *rac*, could explain the difference between our "MG1655" and MDS42, which is a derivative of MG1655. We obtained our "wild-type" MG1655 strain (that we call IA227) from the Silhavy lab and assumed that it is golden. However, we found out that Rof expression from a plasmid does not strongly inhibit growth of either *E. coli* B (BL21) strain or the Keio collection parent strain (BW25113), and Susan Gottesman's lab observed the lack of Rof toxicity in K12 strains. When we sequenced the genome of IA227, we found a number of mutations, including early stop codons in *rpoS* and *glpD*, nonsynonymous codon changes in *spoT*, *folD*, and *hslU*, and IS insertions into *dgcJ* and *relA*, among others. But the prophages are shared by the two strains. Many IA227-specific changes could lead to enhanced Rof sensitivity; the *rpoS* null is one of them (see below and response to point 3).

That said, we expect that induction of many silenced xenogenes, including prophages, makes Rof expression toxic; consistently, our data show that the deletion of *hns* makes *E. coli* hypersensitive to Rof expression. However, the Rof-sensitive phenotype appears to be quite multifactorial. We found that fixing of the *folD* gene in IA227 conferred partial resistance to Rof expression; we hypothesize that FolD could affect Rof synthesis/stability because our MS analysis showed that Rof is acetylated on the N-terminal Met. In support of the importance of Rof modification, a deletion of an uncharacterized acetyltransferase confers sensitivity to Rof.

Our preliminary transcriptomic data show that the overexpression of wildtype Rof causes a large-scale disruption in gene expression, with nearly a quarter of all genes experiencing an increase of at least two-fold \log_2 fold change compared to cells expressing the inactive Rof Y13A variant. While some changes may not be directly caused by Rof-mediated inhibition of ρ , many genes activated by Rof are also activated upon BCM treatment. The most strongly induced gene ($\log_2=27$) is *hokC*, which encodes a member of a conserved family of Hok toxins.

HokC is known to trigger cell growth arrest and rapid death and is a possible reason why *E. coli* dies when Rof is overexpressed. However, we realize that the overexpression toxicity assay, although suitable for testing of the effects of individual Rof residues on Rho inhibition, is not the right way to study the cell physiology. We are currently looking for ways to investigate the cellular function of Rof under physiological conditions.

We revised the manuscript as follows: We deleted original Supplementary Figure 1a and the paragraph in the Results section that referred to these data. We also revised the text to point out the differences between our “wild-type” strain and MG1655 and included the list of differences in the revised manuscript as new Supplementary Table 1. Revised text (line 107):

By contrast, Rof expression was not toxic to wild-type *E. coli* and *Salmonella typhimurium* strains.^{23,30} To determine the cause of Rof toxicity in IA227, we sequenced its genome; we found that IA227 differs from the MG1655 reference genome in positions of mobile elements and several nucleotide polymorphisms (Table S1). Among them is an early stop codon in *rpoS*, which encodes a “general stress response” σ^S factor that orchestrates adaptation to changes in cellular environment and adverse conditions³¹. Our results show that the *rpoS* null sensitizes cells to Rof (Fig. S1b), suggesting an interplay between Rof activity and (still unknown) stress.

We also found that fixing of the *foiD* gene in IA227 confers partial resistance to Rof. We hypothesize that FoID could affect Rof synthesis/stability because our MS analysis showed that Rof is acetylated on the N-terminal Met. Furthermore, we found that a deletion of an uncharacterized acetyltransferase confers sensitivity to Rof. Thus, the Rof-sensitive phenotype appears to be multifactorial and we are actively working on sorting this out.

And what are the survivors Rop- or Kil resistant?

Most cells that grow after a prolonged lag phase caused by Rof overexpression reproduce the same delayed growth characteristics in sequential 24-hour growth assays. For example, in a representative sequential growth curve assay, 9 out of 10 biological replicates displayed the same extended lag phase on the second day as the first, and one suppressor mutant arose that behaved like the strain transformed with an empty vector. To better understand this phenotype, we sequenced genomes of four suppressor variants. All suppressors had an internal deletion of *pcnB*, which would be expected to reduce the copy number of $P_{trc-rof}$ plasmid, and thus Rof expression. In addition, the suppressors each had a range of other mutations throughout the chromosome, suggesting the bacteria were undergoing stress-induced mutagenesis to cope with Rof overexpression. In our previous genetic selection experiments in a derivative of IA227, the *rac* prophage was frequently lost, but we did not observe changes in *rac* in Rof challenge survivors. We are currently testing contribution of different factors induced by Rof to cell survival.

2. From the Discussion “Under stress conditions when dedicated anti-termination machineries and translating ribosomes are scarce, ρ action may need to be tuned down“. Why? With fewer ribosomes, one would want less mRNA, not more. Also, where is the evidence that Rop is more active or more essential during stress?

Rationalizing cellular responses, even in *E. coli*, is challenging, thus we are basing this hypothesis on the available experimental evidence. Why do we think that ρ needs to be inhibited during stress? YihE, the only other known cellular inhibitor of ρ , is expressed during Cpx envelope stress response and proteotoxic stress (ethanol) is relieved by a partial loss-of-function mutation in *rho*. Rof structural homolog Hfq also inhibits Rho, likely indirectly, and is linked to many stresses. We revised the discussion to make this clearer (line 405):

A deleterious mutation in *rho* enables *E. coli* survival during proteotoxic stress⁵⁰ and envelope stress turns on the expression of YihE, a Ser/Thr kinase that inhibits ρ /RNA interactions²⁰. These findings suggest that ρ activity may need to be tuned down under some stress conditions. The RNA chaperone Hfq, which has been implicated in diverse stress responses⁵¹, inhibits ρ function^{16,19} as does another Sm-like protein, Rof²⁵, but their mechanisms remained incompletely understood. Here we present structural, functional and phylogenetic analyses of ρ regulation by *E. coli* Rof, which are in agreement with a study of *Salmonella* Rof by Zhang et al.³⁰.

We do not know which stress conditions lead to increased Rof expression from the chromosome in *E. coli*. The first study that identified Rof as a ρ inhibitor showed that Rof is expressed during transition to the stationary phase. The available proteomic data are inconclusive and we do not yet have specific anti-Rof antibodies (our first attempt failed).

3. “the rof deletion in E. coli has no known growth phenotypes“ Which phenotypes – response to stress?

There are no published data on any phenotypes of *rof* deletion in *E. coli*, but Rof has been reported to promote expression of *Salmonella* SPI-1 (Zhang et al., 2023; reference 30 in the revised manuscript; the accompanying paper), which is silenced by ρ .

We ran a Biolog screen to identify conditions that make Rof beneficial for *E. coli* growth (by comparing the wild-type IA227 and a deletion of *rof* therein). We had a number of hits that were not confirmed with compounds acquired from commercial sources, but two were, amino acid analogs L-norvaline and L-homoarginine. Both are known substrates for aminoacyl-tRNA synthetases and are incorporated into proteins; norvaline was used to induce stringent response in *B. subtilis* (PMID: 11948165) and is produced during oxygen limitation in *E. coli* (PMID: 18940002). Both norvaline and homoarginine preferentially inhibit growth of the Δrof strain, but the effects are not dramatic. We need to collect more data and repeat phenotypic screens before these results can be made public.

Does overproduction of Rof slow the growth of stressed E. coli?

Our result that the *rpoS* deletion makes *E. coli* more sensitive to Rof overexpression is consistent with a view that excess Rof is more deleterious during stress. We added the effects of the *rpoS* deletion as Supplementary Fig. 1b and the following text to the revised manuscript (line 456):

Consistently, we found that the absence of σ^S , a master regulator of general stress response, sensitizes *E. coli* to Rof expression (Fig. S1b).

Narrowing this effect down to particular stress(es) needs work, which we are in the process of doing. We think that most of the additional data described in responses to points 1-3 are very preliminary and are tangential for the understanding of the antitermination mechanism of Rof, i.e. the focus of the present manuscript, and we therefore refrained from including them in the present manuscript.

4. The Discussion details two competing models of Rho-mediated termination. Does the present work favor one of these?

Results presented here cannot distinguish if the observed effects of Rof on termination are based on either of the two pathways nor can they favor one. Rof sequesters the ρ PBSEs that are important for the initial recognition and binding of the *rut* site in the RNA, a key first step in

the RNA-dependent model of ρ termination and a prerequisite for subsequent loading of RNA to the SBS, ring closure and translocation. With respect to the RNAP-dependent model of ρ termination, Rof binding at the ρ PBSes hinders docking of ρ to the elongation complex. As Rof depletes the available number of free ρ hexamers for binding to either, RNA or elongation complex, it would equally affect both mechanisms of ρ -dependent termination. We clarified this in the revised text (line 450):

As Rof binding at the ρ PBSes would, thus, equally affect both, the RNA-dependent and EC-dependent termination pathways, our results do not favor one over the other.

We also would like to point out that the two model for ρ -dependent termination are not necessarily “competing” in the sense of being mutually exclusive. As pointed out in Said et al., 2021 (reference 33 of the revised manuscript), it is also possible that in vivo, EC-dependent steps are a prelude to ultimate termination according to the classical mechanism. We also now point this out in the revised manuscript (line 400):

In the EC-dependent pathway, which may also represent a prelude to ultimate termination via the classical mechanism³³, ρ associates with and accompanies ECs without immediately terminating transcription⁴⁹; upon EC pausing, NusA and NusG can cooperate with ρ in inducing conformational changes in RNAP that stall transcription before ρ engages the transcript^{33,34}.

5. Are Rof deletions synthetic lethal to YihE deletions?

No, we tested this and the double deletion grows well. We think that these two proteins respond to different triggers.

6. “Rof itself is a PBS ligand” Is this incompatible with the statement: “bound Rof would block RNA binding at the extended PBS (Fig. 5e)”? Why then would it have to be bound?

We agree with the reviewer that the term “PBS ligand” is misleading in the context of the inhibitory function of Rof and changed it in the corresponding text (line 425):

Furthermore, as Rof itself binds at the ρ PBSes – why does Rof binding not induce ring closure?

7. Figure 9 (Model) is not instructive; there is no panel showing Rof bound to anything.

We disagree with this reviewer on this point. The reviewer probably oversaw that in Fig. 9 (bottom panel) the schematic shows clearly five Rof molecules bound to one ρ hexamer in agreement with a structure determined in this study.

Reviewer #2 (Remarks to the Author):

The authors report a cryo-EM structure of transcription termination factor Rho in complex with the transcription antitermination factor Rof. The structure shows that Rof inhibits Rho function by occluding the Rho primary binding site (PBS) for RNA and by interfering with formation of the catalytically competent, closed-ring conformational state of Rho. The manuscript will be of interest to researchers in bacterial transcription and transcriptional regulation and should be acceptable for publication in Nature Communications after revision to credit published results and to differentiate between new and confirmatory results.

We thank the reviewer for considering our manuscript to be of interest to a broad community of researchers and in principle acceptable for publication in *Nat Commun*, provided adequate revision.

Specific comments:

lines 79-82: Delete sentences starting with "To further..." and "This structure revealed." The results described in these two sentences confirm results reported in ref. 47. They do not "reveal" anything.

We agree with this reviewer that our ρ -*rut* RNA structure confirms findings in Molodtsov et al., 2023 (reference 24 in the revised manuscript) concerning the extended PBS and have changed the text accordingly (line 80):

Our structure of ρ bound to a 99-nt long RNA that concomitantly occupies the PBSes and the SBS confirmed the existence of an extended PBS²⁴ that is blocked upon Rof binding.

lines 243-289: Shorten this section to one short paragraph. (Results in this section are strictly confirmatory.) Start the paragraph with a sentence stating that ref. 47 reported structures of Rho pre-termination complexes that defined the full interaction between Rho PBS and RNA., and continue with a sentence stating that the authors obtained a structure that confirms the interactions between the Rho PBS and *rut* RNA reported in ref 47.

We agree with the reviewer that the structure of a pre-termination complex reported in Molodtsov et al., 2023 (reference 24 in the revised manuscript) is certainly consistent with RNA bound along an extended region of the ρ NTD. However, given the limited local resolution of the corresponding reconstruction in the region of ρ and bound RNA, we do not think that the structure "defined the full interaction between ρ PBS and RNA" in the sense of revealing detailed interactions or in defining the exact path of the RNA along the PBSes. We also note that in contrast to the ρ -*rut* RNA structure reported here, in which PBS-binding regions and SBS-binding regions are contained on the same RNA molecule, a pre-termination complex reported in Molodtsov et al., 2023 (reference 24 in the revised manuscript) was reconstituted with two different RNA molecules engaging the ρ SBS and PBSes. We also believe that our analyses are not only confirmatory, as we also conducted targeted mutagenesis and, thus, validated the importance of ρ residues lining the extended PBS for RNA binding. We, therefore, changed and shortened the corresponding text passage as follows (line 247):

However, a recent structure of a pre-termination complex showed that RNA can additionally occupy a region proximal to the core PBS²⁴, consistent with the observation that longer PBS ligands bind ρ more tightly and have greater impact on ρ ring closure^{3,6}. We attempted to further narrow down the path of RNA along the ρ PBSes by determining a cryoEM/SPA structure of ρ in complex with a 99-nt long, natural *rut* RNA derived from the λ *tR'* terminator, which encompasses both SBS- and PBS-binding regions (Fig. S7a). In the presence of ADP-BeF₃, we obtained one reconstruction at a global resolution of 2.9 Å (Fig. S8 and Fig. S9; Table S2).

In the ρ -*rut* RNA structure, ρ adopts a closed conformation very similar to ρ in complex with an rU₇ SBS ligand and ADP-BeF₃ (PDB ID: 5JJI³) and to ρ in a pre-termination complex (PDB ID: 8E6W²⁴; Fig. 5c). Density for six nucleotides (nts) is clearly defined at the SBS in the center of the ring (Fig. S7b). We tentatively assigned the sequence at the very 3'-end of the RNA ligand to this region (Fig. S7a). Density corresponding to RNA regions at the ρ PBSes is fragmented, suggesting that corresponding RNA regions are dynamically bound and precluding their reliable modeling. Irrespectively, the reconstruction is consistent with six nts bound in an extended conformation (Fig. 5d). No density is observed connecting the PBS-bound RNA regions or between a PBS-bound RNA region and RNA at the SBS (Fig. 5c). At

the PBSes, the two 3'-most nts are accommodated at the core PBS as observed before³. Density for the preceding nts is lined by ρ^{K102} , ρ^{R105} and ρ^{K115} on one side, and by ρ^{D60} , ρ^{F62} , ρ^{P83} , ρ^{S84} , ρ^{Q85} , ρ^{R87} , ρ^{R88} and ρ^{F89} on the other (Fig. 5e). Consistent with this observation, binding of ρ^{R88E} and ρ^{K115E} variants to the λ *tR' rut* RNA was completely abolished or significantly reduced, respectively, compared to ρ^{wt} , while binding of ρ^{F89S} was undisturbed (Fig. S7c). These results confirm and further validate an "extended PBS" as observed in a pre-termination complex²⁴, which augments the RNA affinity and specificity of the ρ core PBS.

line 296: Start paragraph with sentence stating "The classical [cite reviews] and recently confirmed [cite ref. 47] mechanism of Rho-dependent termination involves ATP-hydrolysis-dependent 5'-to-3' translocation of Rho on RNA."

We adjusted the text to read (line 277):

The classical mechanism of ρ -dependent termination³⁹, recently visualized by cryoEM²⁴, involves ATP hydrolysis-dependent 5'-to-3' translocation of ρ on RNA.

lines 314-316: Delete sentence starting with "However"; replace "in our Rho-rut structure" by "in the structure of the Rho pre-termination complex of ref. 47 and in our Rho-rut structure"; and replace "solve this riddle" by "explain this phenomenon."

Done. The sentence now reads (line 297):

RNA binding at the extended PBS observed in a pre-termination complex²⁴ and in our ρ -*rut* structure could explain this phenomenon.

lines 320-325: Delete.

Done.

line 417: Replace "In vitro" by "It has been hypothesized that."

The manuscripts we refer to describe single-molecule fluorescence assays monitoring ρ -dependent termination in vitro. The observations are consistent with the idea that termination can proceed via either of the two pathways. We therefore adjusted the text to (line 394):

Single-molecule fluorescence studies are consistent with the idea that ρ -dependent termination can proceed via two pathways.^{46,47}

line 419: Insert "classical" before "RNA-dependent."

Done.

line 420: Insert "hypothesized" before "EC-dependent."

Both pathways have been "confirmed" by cryoEM analyses. As pointed out above, other pieces of evidence also are consistent with the idea that ρ can resort to either of the two pathways, at least in vitro. Presently, it remains to be determined, which of the two pathways is preferred in vivo. Indeed, it is even possible that EC-dependent steps are a prelude to final termination according to the classical mechanism, as also pointed out in Said et al., 2021 (reference 33 of

the revised manuscript). We, therefore, believe that it is not justified to contrast the two models as “classical” on the one hand and as “hypothesized” on the other. We thus changed the text to (line 400):

In the EC-dependent pathway, which may also represent a prelude to ultimate termination via the classical mechanism³³, ...

lines 439-441: Replace "in a manner similar to our Rho-rut structure" by "as confirmed by our Rho-rut structure," and delete "; however (Fig 5d)."

Done. The sentence now reads (line 418):

A recent structure of a ρ pre-termination complex²⁴ disclosed RNA bound along extended PBSes and at the SBS of a closed ρ ring, as confirmed by our ρ -rut RNA structure.

Reviewer #3 (Remarks to the Author):

Said et al reported the cryo-EM structure of ρ /Rof and ρ /rut RNA complexes describing the antitermination mechanism driven by Rof which has not been well characterized before. Rof is an antitermination factor and binds to the PBSes site and blocks PBS/RNA interactions, preventing ρ from terminating the transcription. Moreover, the authors suggest that during the EC-dependent termination events, Rof also competes with NusA/NusG/EC complex preventing the assembly of pre-termination ρ /NusA/NusG/EC complex as an antitermination mechanism. This well-rounded article describes the structural details of ρ /Rof, and ρ /rut RNA interactions and proposes the Rof-dependent mechanisms of antitermination. The authors support the proposed mechanism with solid biochemical and genetic pieces of evidence and also imply that Rof-driven antitermination may have differences within Vibrionaceae in comparison to Enterobacteriaceae. There are several concerns, especially those on the cryo-EM maps and coordinates that need to be addressed.

We thank the reviewer for considering our manuscript well-rounded and our biochemical and genetic work as solid.

Major Concerns:

1. About the cryo-EM maps and coordinates.

(1) Rho5-ADP-Rof4: The densities of 5 copies of Rho are good, but those of Rof are poor. 4 ADP molecules were built. Three ones have density while the one on chain E has very poor density, which should be removed from the model. The densities of Rof on the Rho-Rof interfaces are poor (almost no side chain densities, ~6Å). Those need to be clarified in the text. The quality of the density doesn't look like the claimed 2.7Å. A further local refinement on Rof may be useful for improving the density.

(2) Rho6-ADP-Rof5, Rho5-Rof4, and Rho6-Rof5: Same as what has been observed in Rho5-ADP-Rof4 structure: good density of Rho, local resolution of Rof is low (~5-6Å).

We agree with the reviewer that different Rof molecules and nucleotides are differently well defined in the reconstructions. We removed poorly defined ADPs (on chain E in the ρ_5 -ADP-Rof₄ structure, on chain B in the ρ_6 -ADP-Rof₅ structure) from the models. New PDB files and validation reports are provided.

The global resolutions of the reconstructions that we quote are derived from gold-standard refinement, as is common practice. Local resolutions of the reconstructions indeed vary, as is typical for cryoEM structures, with lower local resolution in regions corresponding to Rof molecules and parts of terminal ρ subunits. To better illustrate this feature, we adjusted the coloring scheme in the local resolution plots in Supplementary Figs. 2, 4 and 8 (now panels e). We also now point out this situation in the revised text (please see below).

We built an initial atomic model of ρ -bound Rof based on the density for Rof_c in the ρ_6 -ADP-Rof₅ structure, which is the best defined of any Rof molecule, and for which specific side chain contacts between ρ and Rof could be reliably modeled (please also see below). The resulting model was used as a starting model for all other Rof molecules. Based on the reviewer's suggestion, we conducted a local refinement (for Rof_c of the ρ_6 -ADP-Rof₅ complex); however, this did not further improve the reconstruction in the relevant region.

The reconstructions do not reveal obvious differences in the binding modes of different Rof molecules. We now point out that our discussion of ρ -Rof interaction details refers to the best defined Rof molecule, Rof_c, of the ρ_6 -ADP-Rof₅ structure (please see below).

We now describe the above aspects in the revised text (line 141):

The local resolutions of the reconstructions decrease towards the regions corresponding to terminal ρ subunits and bound Rof (Fig. S2e and Fig. S4e). Weaker density for the peripheral Rof molecules could indicate a higher flexibility or a lower occupancy. We built an initial atomic model of ρ -bound Rof based on the density for Rof_c in the ρ_6 -ADP-Rof₅ structure, which is best defined (Fig. 2b,c). This model was used as a starting model for other Rof molecules in the various complexes. The reconstructions do not reveal differences in the ρ -Rof interfaces, irrespective of oligomeric state or bound nucleotide. We therefore focus our subsequent descriptions on the ρ_6 -ADP-Rof₅ complex and on Rof_c when discussing interaction details.

Also, we now include density in revised Fig. 2d-h, showing that side chain conformations could be reliably modeled at the interface of the best defined Rof_c of the ρ_6 -ADP-Rof₅ complex:

d-h, Details of the ρ -Rof interaction. Interacting residues are shown as sticks with atoms colored by type; carbon, as the respective protein subunit; nitrogen, blue; oxygen, red. Black dashed lines, hydrogen bonds or salt bridges. Regions of the ρ_6 -ADP-Rof₅ cryoEM reconstruction are shown as semi-transparent surfaces.

(3) Rho-rutRNA: very good density and models in most parts. Those of ADP are clear. The part of RNA bound in the center of hexameric Rho is clear, but the claimed part bound around Chain C Tyr80 (PBS) is relatively poor, it's impossible to build more than 2nt RNA residues. In addition, the RNA sequence in the Rho center (SBS) might refer to the previous structure (the author didn't clearly state how to build RNA in the methods), but it's impossible to determine what sequence should be placed around chain C Tyr80 (PBS). Authors should remove the unreliable RNA part and provide additional evidence to support how and why they build such RNA sequences here.

We agree and have revised our descriptions accordingly. Regarding sequence for the well-defined SBS-bound RNA, we now clarify (line 258):

We tentatively assigned the sequence at the very 3'-end of the RNA ligand to this region (Fig. S7a).

Regarding RNA at the PBSes, we fully agree that sequence or even the precise conformation or specific contacts to ρ cannot be modeled reliably. Indeed, we refrained from modeling RNA at the PBSes except for one PBS where density is best defined. Again, we conducted a local refinement, which did not improve the local density. However, the density at the PBSes is consistent with six nucleotides bound in an extended conformation and covering an extended PBS. By showing an RNA model at the PBS, we mostly wanted to illustrate that the RNA-related density at the PBSes covers an "extended PBS" and defines the approximate path of the RNA along this extended PBS. We further validated the importance of ρ residues at the extended PBS for RNA binding by site-directed mutagenesis in combination with SEC-based interaction assays. For insights into how Rof interferes with RNA binding at the PBSes, the precise sequence of the PBS-bound RNA regions is irrelevant and we removed all descriptions referring to this sequence from the revised text. In addition, we asked the Protein Data Bank to add a REMARK to the submitted PDB file, indicating that corresponding RNA residues cannot be modeled reliably. Our more careful descriptions in the revised manuscript now read (line 259):

Density corresponding to RNA regions at the ρ PBSes is fragmented, suggesting that corresponding RNA regions are dynamically bound and precluding their reliable modeling. Irrespectively, the reconstruction is consistent with six nts bound in an extended conformation (Fig. 5d). No density is observed connecting the PBS-bound RNA regions or between a PBS-bound RNA region and RNA at the SBS (Fig. 5c). At the PBSes, the two 3'-most nts are accommodated at the core PBS as observed before³. Density for the preceding nts is lined by ρ^{K102} , ρ^{R105} and ρ^{K115} on one side, and by ρ^{D60} , ρ^{F62} , ρ^{P83} , ρ^{S84} , ρ^{Q85} , ρ^{R87} , ρ^{R88} and ρ^{F89} on the other (Fig. 5e). Consistent with this observation, binding of ρ^{R88E} and ρ^{K115E} variants to the λ *tR' rut* RNA was completely abolished or significantly reduced, respectively, compared to ρ^{wt} , while binding of ρ^{F89S} was undisturbed (Fig. S7c). These results confirm and further validate an "extended PBS" as observed in a pre-termination complex²⁴, which augments the RNA affinity and specificity of the ρ core PBS.

We also removed Figure panels in which we had suggested specific contacts between PBS-bound RNA and ρ .

(4) Figures of model-to-map FSCs need to be present to demonstrate the quality of the models and the corresponding ones (model resolution when FSC=0.5) should also be included in Table S2. The accuracy of the model is important for structural explanations and descriptions.

We have included model-to-map cross resolutions (FSC 0.5) as determined during phenix real space refinement in Supplementary Table 2. Corresponding FSC plots are now included in Fig. S2, S4 and S8 (new panels c). Below, a collection of the new panels:

(5) Additional figures showing the density of different parts of the complex should be included to justify the claimed 3.3-2.7 resolutions.

We now include density in revised Fig. 2d-h (please see above), showing that side chain conformations could be reliably modeled at the interface of the best defined Rof_c of the ρ_6 -ADP-Rof₅ complex. We also included density for the SBS-bound RNA in revised Fig. S7b:

b
b, ρ binds SBS RNA via the Q- and R-loops. RNA, Q-loop residues (281-287) and R-loop K326 of the ρ subunits are depicted as sticks. The RNA oriented with the 5'-end at the bottom and the 3'-end at the top. The region of the cryoEM reconstruction covering the six nucleotides at the center of the ρ ring is shown as a semi-transparent surface. Two/one residue/s 5'/3' of this region are additionally defined by weaker density, not visible at the present contour level. The three panels show the three pairs of ρ subunits positioned opposite to each other around the RNA. Most of the contacts are not sequence-specific and are mediated by backbone interactions of both, RNA and ρ .

Density for PBS-bound RNA is shown in Fig. 5d as before:

d
(6) It is hard to understand the reported resolutions in the “Resolution (A)” part in Table S2. Please follow the general format shown in other cryo-EM papers.

We thank the reviewer for catching this issue and have corrected the Supplementary Table 2 accordingly.

2. In the text, authors described and showed the main interaction on the Rho-Rof interface, which is involved in 7 residues of Rof, but they only performed detailed mutagenesis and functional studies on part of them. In some places of the text, they only emphasized three, but also discussed other residues in other places. All these make reading confusing. The authors should clarify the logic of why and how they chose residues to do the mutagenesis and functional studies.

We thank the reviewer for this comment and have tried to clarify the situation in the revised text (please see below). In addition, we now systematically refer to the Fig. 2d-h panels that show interaction details involving the exchanged residues.

We prepared a limited number of Rof variants for interaction testing with ρ via SEC (Y13A, D14A and E17A). This set of experiments was complemented by testing of ρ variants (R88E, F89S, K115E). We now mention that the targeted Rof residues reside in helix $\alpha 1$ (line 176):

To validate our structural observations, we replaced Rof residues Y13, D14 or E17 in helix $\alpha 1$ with alanine and tested binding of the corresponding Rof variants to ρ^{wt} in analytical SEC (Fig. 3a).

We then tested the same set of Rof variants (Y13A, D14A, E17A) in in vitro anti-termination assays. Finally, we tested the effect of exchanging Y13, D14, E17 and, additionally, C10 of Rof on cell toxicity. We now clearly state this in the revised text (line 191):

Next, we evaluated the effects of exchanging Y13, D14 and E17 and an additional ρ -contacting residue in helix $\alpha 1$ of Rof, C10, on toxicity in vivo.

Instead of the E17A variant, which did not exhibit a significant defect in ρ binding, we tested the putatively more severe, charge-reversed E17R, as explained already in the original text (now line 197):

As Rof^{E17A} did not show significant defects in vitro (Fig. 3b), we tested the effects of the Rof^{E17K} variant instead; ...

We then decided to also test effects of residue exchanges in another ρ -contacting region, the $\beta 3$ - $\beta 4$ loop, on cell toxicity (R46A, K47A and N48A). Again, we clearly state this in the text (now line 199):

By contrast, substitutions in the Rof $\beta 3$ - $\beta 4$ loop (Fig. 2g) had smaller effects; ...

We hope that with these explanations and additions to the text, our rationale and the targeted residues are easier to follow.

3. Further kinetic studies may be necessary to confirm the competition between EC/NusA/NusG/ ρ and EC/NusA/NusG/ ρ +Rof suggested by Figure 7 and support the claim that Rof also competes with NusA/NusG/EC complex preventing the assembly of pre-termination ρ /NusA/NusG/EC complex.

We agree with the reviewer that the results shown in Fig. 7b of the original manuscript were not entirely convincing. We therefore repeated the experiments with freshly prepared components. In addition, we now used a short, 16-nt RNA that does not expose binding regions outside the RNA exit channel (as also used for reconstitution of a ρ -bound EC in Hao et al, 2021; reference 34 in the revised manuscript); in this manner, we monitored binding of ρ to the NusA/NusG-EC independent of possible ρ -RNA interactions. Under these conditions, we obtained a clear-cut picture: pre-bound Rof prevents ρ from engaging the NusA/NusG-EC (compare panels 2 and 3) and Rof added in excess to a pre-formed ρ /NusA/NusG-EC displaced ρ (compare panels 2 and 4). Here the revised Fig. 7b:

b, SDS-PAGE analysis of SEC runs, monitoring ρ binding to ECs in the absence and presence of Rof. First panel, NusA/NusG-EC. Second panel, pre-formed NusA/NusG-EC incubated with a three-fold molar excess of ρ hexamer. Third panel, pre-formed NusA/NusG-EC incubated with a three-fold molar excess (relative to ρ hexamer) of ρ -Rof complex. Fourth panel, pre-formed ρ /NusA/NusG-EC incubated with a ten-fold molar excess of Rof (relative to ρ hexamer).

Minors:

1. lines 320-325 could be an overstatement unless confirmed by improving density. “It is also conceivable, albeit not resolved in our ρ -rut RNA 320 structure, that the 5'-end of RNA regions bound at the extended PBS directly contacts the 321 neighboring connector helix α_5 . PBS-bound RNA could thereby stabilize the connector in the 322 closed-ring configuration and support an apparent rotational movement of the NTD, which 323 facilitates displacement of the three-helix bundle from the neighboring NTD, thus supporting 324 retraction of ρ R28 from its inter-subunit binding pocket (Fig. 6a,b).” May be moved to the discussion section instead as a logical assumption.

We deleted this sentence as the speculation is not required for our conclusions.

2. Figure 8D may need additional modification (clearer label or ecRof and maybe an additional panel with actual superposition of the monomers (not dimer) to confirm the statement in the main text).

We apologize, the original legend of Fig. 8d was misleading. We indeed superimposed vcRof on ecRof from the ρ -Rof structure but omitted ecRof in this figure. We modified Fig. 8d as follows:

d, Superposition of the *vcRof* on *ecRof* bound to ρ . Upper panel, superposition of a *vcRof* monomer on *ecRof* bound to ρ . Lower panel, superposition of the *vcRof* dimer on *ecRof* bound to ρ ; for clarity, *ecRof* is not shown. Rotation symbols, view relative to Fig. 2b. While monomeric *vcRof* would align without steric conflict, there might be steric hindrance in the interaction between a dimer of *vcRof* and ρ .

3. A new panel of superimposition in Figure 4 is necessary to show what conformational change is observed upon Rho binding.

We agree and added a panel to Fig. 4 (new panel c) that shows a superposition of isolated *ecRof* with ρ -bound *ecRof* to better illustrate the conformational changes that occur upon ρ binding:

c, Isolated *ecRof* (gray) superimposed on ρ -bound *ecRof* (violet) in the same view as isolated *ecRof* in (a). Rotation symbols, view relative to Fig. 2b. In the conformation of isolated *ecRof*, the N-terminus and helix α_1 would sterically interfere with binding to ρ .

Reviewer #4 (Remarks to the Author):

In the manuscript titled "Sm-like protein Rof Inhibits Transcription Factor ρ by Binding Site Obstruction and Conformational Insulation," Said et al. demonstrated the molecular mechanism of anti-termination by Rof using cryo-EM structures of the Rof-bound ρ complex. *E. coli* Rof (or YaeO) is a 9-kDa acidic protein whose expression is regulated by the growth rate. It stably binds ρ and reduces ρ -dependent termination. While its anti-termination activity *in vivo* was known, its *in vitro* activity was not confirmed. To gain a molecular understanding of how Rof inhibits ρ , the authors initially found that N-terminal histidine tagging negates Rof's activity by examining the effects of Rof variants on cell growth. Subsequently, the authors demonstrated that tag-cleaved Rof indeed inhibits ρ -dependent termination using an *in vitro* transcription assay and solved the cryo-EM structure of the ρ -Rof complex. In order to compare the Rof-bound ρ structure with a native structure, the team also solved the ρ -rut RNA complex structure. Although pre-termination complex structures containing ρ , rut RNA, NusG, NusA, and RNAP elongation complexes are known and were published by the same group, this ρ -RNA structure was crucial for a precise analysis of Rof's effect on ρ .

In the ρ -Rof complex structure, the authors discovered that the ρ hexamer of an open state is bound to four or five Rof molecules. Each ρ protomer binds one Rof, though the

ρ protomers at the opening displayed relatively weak or no Rof density. The authors demonstrated that a single mutation of the Rof residues responsible for hydrogen bonds or salt bridges at the ρ -Rof interface results in the disruption of ρ -Rof complex formation using analytical SEC, in vitro transcription assay, and growth rate measurement. Rof exhibited significant conformational changes upon ρ binding when compared to its previously resolved crystal structure.

Next, the authors illustrated the competition between Rof and rutRNA for binding to ρ through the use of the ρ -RNA complex structure and fluorescence anisotropy. Furthermore, they revealed that Rof-bound ρ adopts an open state, while its NTD-CTD connector region resembles that of a closed ρ . This observation suggests that Rof not only competes with substrate RNA, particularly the PBS (primary RNA-binding site) substrate but also induces conformational changes in ρ , disrupting the communication between NTD and CTD. This communication is crucial for ρ 's ability to modify the RNAP elongation complex leading to transcription termination.

In conclusion, this review strongly recommends the publication of this manuscript in Nature Communications. Below, you'll find some additional comments on the manuscript.

We thank the reviewer for the strong support of our manuscript.

Minor points

1. In Fig 6a, the ρ_D and ρ_C were drawn whereas ρ_B and ρ_C were in Fig 6b. Was the choice of protomers with different subscripts intentional? In the figure legend, this reviewer is unsure about the comment in the parentheses, specifically “(ρ_B an ρ_C in the closed hexamer)”.

The reviewer points out an aspect that can easily cause confusion. In essence, ρ subunits are historically labeled in opposite directions in structures of open and closed ρ ; as a consequence, subunit interface ρ_C/ρ_D in open ρ is equivalent to subunit interface ρ_C/ρ_B in closed ρ . While this may sound arbitrary, there is a clear logic behind the labeling. In the structure of open ρ (as first reported in Skordalakes & Berger, 2003 (reference 2 in the revised manuscript) and as also observed in our ρ -Rof complexes), the subunits are arranged as a left-handed helix, with the axes connecting the centers of NTD and CTD of each subunit roughly parallel to the helix axis. Defining the helix direction from NTD to CTD of the ρ subunits yields the labeling of subunits from A to F (as used in our manuscript and now explicitly labeled in revised Fig. 2b). In the structure of fully asymmetric closed ρ in complex with SBS-bound RNA (Thomsen & Berger, 2009; reference 5 in the revised manuscript; as also observed in our ρ -rut RNA structure), the six subunits are arranged around a right-handed helical, single-stranded RNA region at the SBS. Five ρ subunits contact the RNA backbone at the SBS, one does not. Subunit labeling in this conformation was chosen to start from the subunit contacting the 5'-most nucleotide of the SBS-bound RNA (A), via subunits contacting stepwise more 3'-located nucleotides (C-E) to the subunit that does not contact the SBS RNA backbone (F). This rationale yields a subunit labeling that is reversed compared to that of open ρ when the rings are oriented with equivalent surfaces (NTD or CTD faces) to the viewer.

To stick with the traditional subunit labeling but hopefully avoid confusion, we now denote the subunits of open ρ with capital letters (A-F) and the subunits of closed ρ with lower case letters (a-f). We also added a brief explanation to the legend of Fig. 5c (line 943):

As originally proposed^{2,5}, ρ subunits are labeled in opposite directions around open and closed ρ rings (compare to Fig. 2b). We therefore labeled subunits of open ρ with capitals (A-F) and subunits of closed ρ with lower case letters (a-f).

2. In lines 267-273, although the manuscript describing that the boxB structure was used as a reference point for the RNA building, this reviewer still cannot understand how the 6-nt portion of rutA bound at the PBS of ρ could be built. Was there a continuous RNA density observed between the boxB patch and the 6-nt RNA? If so, RNA? Could you provide a confident measurement of the number of bases positioned between the boxB patch and the 6-nt RNA?

Due to the poor quality of the density corresponding to RNA regions bound at the PBSes, due to lack of density connecting the PBS-bound RNA regions and due to lack of density connecting one PBS-bound RNA region with the RNA at the SBS, we indeed cannot reliably model RNA at the PBSes. While density for SBS-bound RNA is very clear, sequence assignment is also tentative; we based it on the sequence at the very 3'-end of the RNA ligand. We had refrained from modeling RNA at the PBSes except for one PBS where density is best defined. Further focused refinement did not improve the density in the relevant region. By showing an RNA model at this region, we mostly wanted to illustrate that the RNA-related density at the PBSes covers an "extended PBS" and defines the approximate path of the RNA along this extended PBS. We further validated the importance of ρ residues at the extended PBS for RNA binding by site-directed mutagenesis in combination with SEC-based interaction assays. For insights into how Rof interferes with RNA binding at the PBSes, the precise sequence of the PBS-bound RNA element is irrelevant and we removed all descriptions referring to this sequence from the revised text. In addition, we added a REMARK to the submitted PDB file, indicating that corresponding RNA residues cannot be modeled reliably. Our more careful descriptions in the revised manuscript now read (line 257):

Density for six nucleotides (nts) is clearly defined at the SBS in the center of the ring (Fig. S7b). We tentatively assigned the sequence at the very 3'-end of the RNA ligand to this region (Fig. S7a). Density corresponding to RNA regions at the ρ PBSes is fragmented, suggesting that corresponding RNA regions are dynamically bound and precluding their reliable modeling. Irrespectively, the reconstruction is consistent with six nts bound in an extended conformation (Fig. 5d). No density is observed connecting the PBS-bound RNA regions or between a PBS-bound RNA region and RNA at the SBS (Fig. 5c). At the PBSes, the two 3'-most nts are accommodated at the core PBS as observed before³. Density for the preceding nts is lined by ρ^{K102} , ρ^{R105} and ρ^{K115} on one side, and by ρ^{D60} , ρ^{F62} , ρ^{P83} , ρ^{S84} , ρ^{Q85} , ρ^{R87} , ρ^{R88} and ρ^{F89} on the other (Fig. 5e). Consistent with this observation, binding of ρ^{R88E} and ρ^{K115E} variants to the λ *tR'* *rut* RNA was completely abolished or significantly reduced, respectively, compared to ρ^{wt} , while binding of ρ^{F89S} was undisturbed (Fig. S7c). These results confirm and further validate an "extended PBS" as observed in a pre-termination complex²⁴, which augments the RNA affinity and specificity of the ρ core PBS.

We also removed Figure panels in which we had suggested specific contacts between PBS-bound RNA and ρ .

REVIEWERS' COMMENTS

Reviewer #1 (Remarks to the Author):

The authors have responded to my critique to my satisfaction. I recommend publication of their revised manuscript.

Reviewer #2 (Remarks to the Author):

The revised manuscript satisfactorily addresses my comments.

Reviewer #3 (Remarks to the Author):

The revised manuscript has a better description and representation. New data has been provided and the authors did a great job in response to the comments. All my concerns have been fully addressed. I recommend it to be published.

Reviewer #4 (Remarks to the Author):

The authors have adequately addressed this reviewer's comments in their revised manuscript.